# ⊙⊙🧩 Pixels Lie, Code Doesn't: Thinking with Visual Programming for "Seemingly Impossible" Multimodal Agentic Reasoning Tasks

## Abstract

To overcome the inherent limitations of Chain-of-Thought (CoT) and to further push the upper bound of multimodal reasoning capabilities, we introduce Thinking with Visual Programming (TVP), where models can iteratively interact with an external code executor to generate, run, and verify both visual and textual agentic operations as part of the reasoning loop. Motivated by the open question of how far Multimodal Large Language Models (MLLMs) still lag behind this paradigm, we introduce **MMR-VIP**, a **M**ulti**M**odal Agentic **R**easoning benchmark built on **V**isual **I**mpossible **P**roblems. We design MMR-VIP with two key principles: (1) We construct a **Difficulty Ladder** grounded in computational complexity theory, structuring tasks from easy problems that can be solved with inherent perception and reasoning, through medium problems that require external computational tools, to hard problems that remain intractable even with programming assistance. (2) We decompose the paradigm of Thinking with Visual Programming into three **Cognitive Skills**, namely **Perception**, **Abstraction**, and **Optimization**, which correspond to perceiving visual inputs, abstracting them into problem formulations, and optimizing algorithms to obtain efficient solutions. Our experiments on MMR-VIP yield the following findings: (1) GPT-5, as a native TVP model, delivers the strongest overall results, yet its accuracy remains only 38.2%, underscoring substantial room for progress. (2) For commercial models, multi-turn code execution consistently surpasses direct CoT and single-turn execution, providing stable and significant improvements. (3) Across difficulty levels, performance follows a ladder-shaped trend, with negligible gains on easy tasks, the largest improvements on medium tasks, and steady advances on hard tasks. (4) From a cognitive perspective, TVP enhances optimization by offloading complex computation, search, and planning, but models still encounter bottlenecks in abstraction.

## 1 Introduction

Multimodal reasoning is a defining capability of human intelligence, enabling us to address diverse challenges such as navigating in the physical world, interpreting scientific figures, and solving geometry problems (Yue et al., 2024; Lu et al., 2024). Recent advances in Multimodal Large Language Models (MLLMs) (OpenAI, 2024; DeepMind, 2025; Bai et al., 2025) have demonstrated significant progress by leveraging **Chain-of-Thought** (CoT) (Wei et al., 2022; Zhang et al., 2024c), which bridges perception and reasoning through explicit sequences of textual steps. Nevertheless, existing improvements remain constrained, since they primarily extend text-based CoT, prolonging the reasoning process without enhancing the upper bound of the model's reasoning capabilities.

To address these limitations, "**Thinking with Images**" (**TWI**) (Zheng et al., 2025; Su et al., 2025a;c) has recently been proposed as an emerging paradigm, enabling models to incorporate iterative visual inspection as part of the reasoning loop. In practice, current methods rely on manipulating images with external tools (Zhang et al., 2025), such as cropping, zooming, rotating, and other image processing operations. Although such strategies enhance perceptual accuracy, they are limited to a narrow set of predefined visual tools, thereby rigidifying the reasoning process and restricting its generality. As various forms of visual manipulation can naturally be expressed and executed through programming, we propose the paradigm of "**Thinking with Visual Programming**" (**TVP**), which goes beyond fixed tools and allows models like OpenAI's o4 and GPT-5 (OpenAI, 2025a) to flexibly generate, execute, and verify both visual and textual operations within the reasoning process.

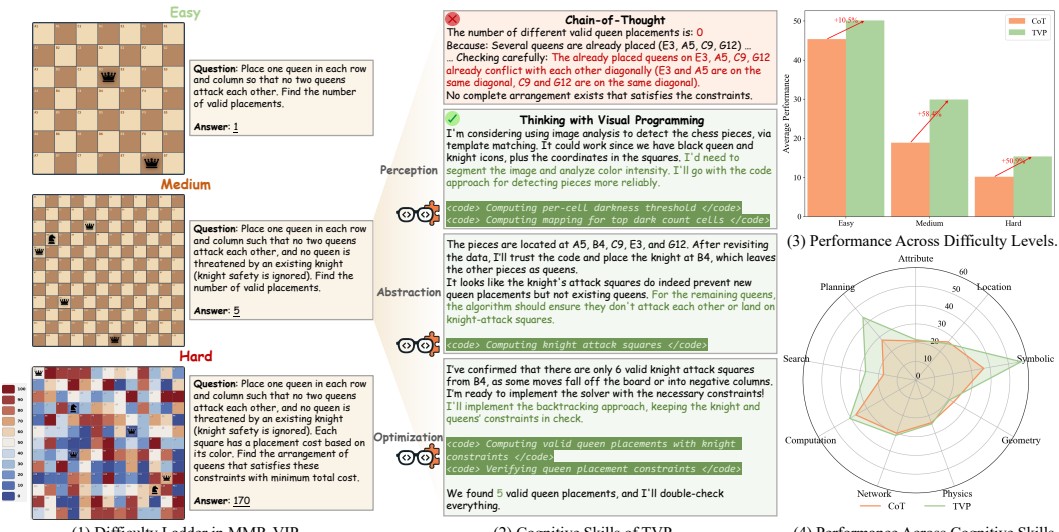

(1) Difficulty Ladder in MMR-VIP.  (2) Cognitive Skills of TVP.  (4) Performance Across Cognitive Skills.

Figure 1: Thinking with Visual Programming paradigm. Figure 1(1) illustrates the three difficulty levels in MMR-VIP using the N-Queens task as an example. Figure 1(2) shows that for a medium-level problem, direct CoT reasoning fails while TVP succeeds, and in the process three key cognitive skills emerge. Figure 1(3) compares the average performance of four powerful models (GPT-4.1-mini, GPT-4.1, Gemini-2.5-Flash, and Claude-Sonnet-4) under CoT and TVP, showing minimal changes on easy tasks, the largest gains on medium tasks, and clear improvements on hard tasks. The results exhibit a ladder-shaped performance trend across difficulty levels. Figure 1(4) presents the performance differences of the four models across cognitive skills, where TVP yields notable improvements in symbolic (perception), computation, search, and planning (optimization).

Humans inherently solve complex reasoning problems in a programming-like manner by preprocessing visual inputs for better perception, applying algorithmic procedures to derive solutions, and verifying outcomes through testing. Nevertheless, it remains unclear how far current MLLMs are from this paradigm. To this end, we introduce **MMR-VIP**, a **M**ulti**M**odal Agentic **R**easoning benchmark that consists of **V**isual **I**mpossible **P**roblems. Formally, we refer to Visual Impossible Problems as problems that appear intractable under CoT-based reasoning, yet become solvable when augmented with visual programming interactions. We design MMR-VIP with two key considerations:

**Difficulty Ladder**. We categorize problems into three levels of difficulty, drawing inspiration from how humans tackle tasks with and without tools, and grounded in computational complexity theory. (1) **Easy** level requires that the model can reliably solve them using its inherent perception and reasoning abilities, without any programming assistance. This level corresponds to "*low-complexity problems in P*", where the model can perform reasoning within its working memory; (2) **Medium** level is challenging for the model to solve independently, but can be effectively addressed when it is allowed to use a code interpreter. This level typically involves "*polynomial-time solvable problems in P*", where the model must rely on external computational tools to compute solutions; (3) **Hard** level remains unsolved even with programming assistance, often due to their large-scale computational complexity, highly intricate constraints, or demanding optimization requirements. This level corresponds conceptually to "*NP-hard problems*", which often lie beyond the capabilities of current models. As shown in Figure 1(1), the three levels form a progressive difficulty ladder, where each step reflects an increasing demand on the model's reasoning capacity and reliance on external tools.

**Cognitive Skill**. We decompose the Thinking with Visual Programming paradigm into three key cognitive skills, focusing on the core cognitive processes required to perceive, abstract, and optimize multimodal agentic reasoning. Taking the N-Queens problem in Figure 1(2) as an example: (1) **Perception** requires the model to transform *visual content* into *structured information*, correctly extracting relevant elements from multimodal inputs (*e.g.*, detecting and locating chess pieces on the board); (2) **Abstraction** requires the model to transform *structured information* to *problem formulation*, producing computationally useful forms and proposing feasible solutions (*e.g.*, converting piece positions into symbolic constraints that capture attack rules); (3) **Optimization** requires the model to transform *problem formulation* to *algorithmic optimization*, optimizing both algorithms

and computational procedures to obtain correct and efficient answers (*e.g.*, applying a backtracking algorithm to search for valid queen placements under the given constraints).

MMR-VIP encompasses **28** carefully crafted task types, each designed across three difficulty levels, resulting in **1,680** instances that provide a comprehensive evaluation of multimodal agentic reasoning capabilities. These tasks span a wide spectrum, from basic skills such as *counting* and *height measurement* to advanced challenges including *graph coloring* and *circuit logic*. To avoid dataset contamination and guarantee that models solve tasks via code execution instead of memorized recall, all problems in MMR-VIP are generated using carefully designed, manually written code.

We conduct a comprehensive evaluation on MMR-VIP across a wide range of MLLMs, including commercial models such as Claude-Sonnet-4, open-source models such as Qwen2.5-VL-72B, as well as native TVP models like o4-mini and GPT-5. We further assess different reasoning paradigms, including direct CoT, single-turn code execution, and multi-turn code execution. We obtain the following conclusions: (1) Our experimental results reveal clear differences across model types and reasoning paradigms. For open-source models, introducing code execution provides little to no improvement, mainly due to their limited visual programming capabilities. For commercial models, single-turn code execution yields unstable performance, while multi-turn code execution consistently delivers substantial gains. As illustrated in Figure 1(3), multi-turn code execution improves accuracy on medium-level tasks by **58.4%** compared to direct CoT. GPT-5, as a native TVP model, achieves the best overall performance; however, its accuracy remains only **38.2%**, indicating substantial room for improvement; (2) Performances across different difficulty levels align well with the design of MMR-VIP, exhibiting a ladder-shaped performance trend. Compared to direct CoT, we observe that TVP yields minimal changes on easy tasks, the largest gains on medium tasks, and consistent improvements on hard tasks; (3) From the perspective of cognitive skills, TVP shows clear progress in optimization, as it can leverage programming to offload complex computation, search, and planning operations. However, its performance still encounters bottlenecks in abstraction, where models lack the ability to translate visual inputs into high-level problem formulations. We hope that MMR-VIP will serve as a challenging benchmark to drive future research toward closing this gap.

## 2 PARADIGM DEFINITIONS

### 2.1 MULTIMODAL CHAIN-OF-THOUGHT

We formalize the conventional paradigm of Multimodal Chain-of-Thought reasoning. For a model $\theta$, given an input image $I$ and a textual question $x$, the CoT process can be defined as:

$$P_{\text{CoT}}(y \mid I, x) = P_\theta(r \mid I, x) \cdot P_\theta(y \mid I, x, r). \tag{1}$$

Here, $r = (s_1, s_2, \ldots, s_n)$ denotes the intermediate reasoning chain, which explicitly captures the sequence of textual steps bridging perception and reasoning, while $y$ represents the final answer conditioned on both the original input $(I, x)$ and the generated textual rationale $r$.

### 2.2 THINKING WITH VISUAL PROGRAMMING

We formalize the proposed paradigm of Thinking with Visual Programming. For a model $\theta$, given an input image $I$ and textual question $x$, TVP extends conventional CoT by introducing programming actions $a_t$, which are executed through interaction with an external code executor $\mathcal{E}$. Unlike single-pass reasoning, this is a multi-turn interactive agentic process consisting of $T$ rounds:

$$P_{\text{TVP}}(y \mid I, x) = \prod_{t=1}^{T} P_\theta(r_t, a_t \mid s_{t-1}) \cdot P_\theta(y \mid s_T). \tag{2}$$

At each step $t$, the model generates a reasoning trace $r_t$ and a programming action $a_t$, executes $a_t$ via the external executor $\mathcal{E}$, and incorporates the multimodal execution result $\mathcal{E}(a_t)$ into the state $s_t$:

$$s_t = s_{t-1} \cup \{r_t, a_t, \mathcal{E}(a_t)\}, \quad s_0 = \{I, x\}. \tag{3}$$

Compared to CoT, TVP offers significant advantages by integrating pixel manipulations and algorithmic computation into the reasoning loop, enabling models to move beyond textual thinking. In this paper, we do not provide models with fixed external tools. Instead, we allow them to write code that can call standard libraries, such as `PIL`, `OpenCV`, and `Matplotlib`, among others.

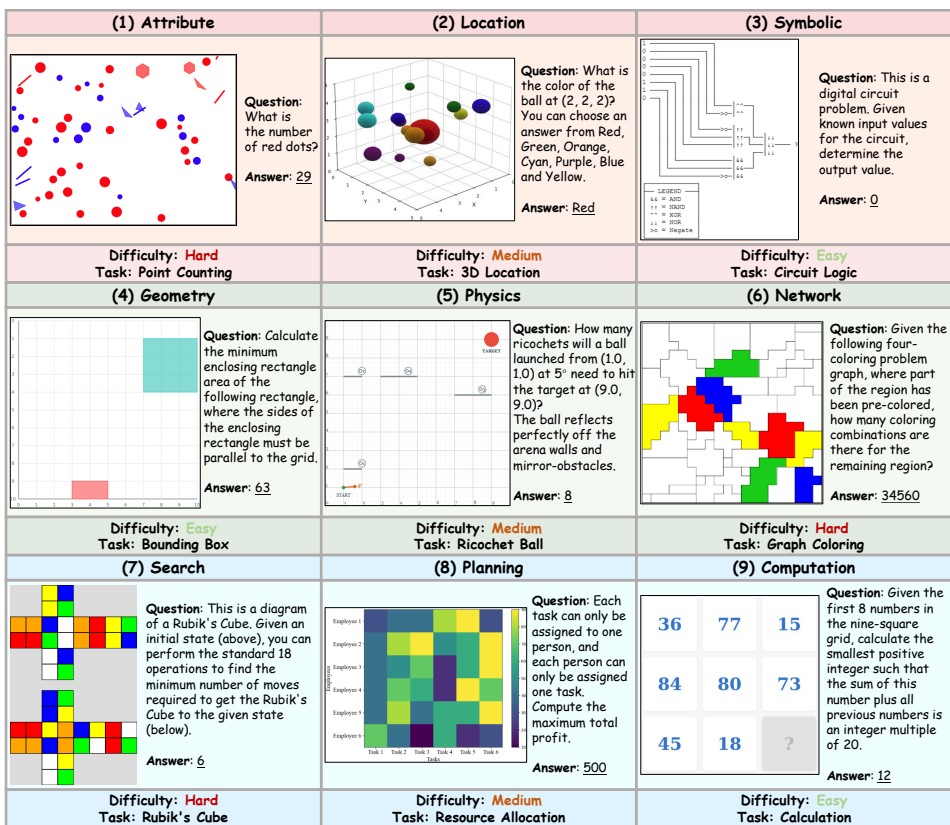

Figure 2: Evaluation framework of cognitive skills in MMR-VIP.

## 3 MMR-VIP BENCHMARK

To investigate how far current MLLMs are from the paradigm of TVP, we introduce **MMR-VIP**, a **M**ulti**M**odal Agentic **R**easoning benchmark that consists of **V**isual **I**mpossible **P**roblems. These are carefully designed problems that existing MLLMs cannot reliably solve with conventional CoT reasoning alone, but instead necessitate interaction with an external code executor. We will detail the design principles behind MMR-VIP, including its difficulty ladder and cognitive skill dimensions, and describe the benchmark construction process along with dataset statistics.

### 3.1 DIFFICULTY LADDER

We categorize problems in MMR-VIP into a three-level **Difficulty Ladder**, drawing inspiration from how humans tackle tasks of varying complexity and their reliance on external tools. At the **Easy** level, tasks can be reliably solved using the model's inherent perception and reasoning abilities, without the need for programming assistance. These correspond to "*low-complexity problems in P*", where solutions can be derived directly within the model's working memory. The **Medium** level encompasses tasks that models struggle to solve on their own but can successfully address when supported by external tools such as code interpreters. These tasks align with "*polynomial-time solvable problems in P*," where deriving solutions requires programmatic operations and computational tools beyond intuition alone. Finally, the **Hard** level captures problems that remain unsolved even with programming assistance, typically due to large-scale computational complexity, intricate constraints, or challenging optimization requirements. Conceptually, these tasks are analogous to "*NP-hard problems*", which often exceed the practical capabilities of current models. Such a difficulty ladder setting enables a more in-depth examination of the paradigm of TVP.[1]

---

[1]The tasks in MMR-VIP are not strictly designed or guaranteed to align with formal complexity-theoretic definitions, but rather follow the spirit of increasing computational and cognitive demands.

Table 1: Mapping between cognitive skills and task types in MMR-VIP.

| Category | Tasks |
|---|---|
| Attribute | 3D Position, Bin Packing, Graph Coloring, Hanoi Tower, Point Counting, Resource Allocation, Rubik's Cube, Sliding Puzzle, Snake Game, Three-Views |
| Location | 3D Position, Bounding Box, Height Measurement, Point Counting, Projectile Motion, Snake Game, Three-Views |
| Symbolic | Calculation, Chart, Circuit Logic, House Robber, Interval DP, N-Puzzle, Projectile Motion, Tableau LP |
| Geometry | Area Measurement, Bounding Box, Rubik's Cube, Three-Views |
| Physics | Circuit Logic, Projectile Motion, Ricochet Ball |
| Network | Graph Coloring, Graph Isomorphism |
| Search | Bin Packing, Bubble Sort, Calculation, Graph Coloring, Maze, N-Puzzle, N-Queens, Path Counting, Rubik's Cube, Sliding Puzzle, Snake Game |
| Planning | Chart, Hanoi Tower, House Robber, Interval DP, Lights Out, Resource Allocation, Tableau LP |
| Computation | Calculation, Path Counting |

## 3.2 Cognitive Skill

Beyond task difficulty, we design MMR-VIP to emphasize the underlying **Cognitive Skills** required for multimodal agentic reasoning under the TVP paradigm. These skills highlight the essential processes through which models must learn to leverage external tools to approach complex problems. We define three successive skills within TVP: **Perception**, **Abstraction**, and **Optimization**, which together examine a model's visual programming ability from complementary dimensions.

**Perception**: This skill concerns the model's ability to accurately extract structured information from raw visual inputs. Unlike direct pattern recognition that relies solely on intrinsic visual perception, TVP enables models to enhance perception through programmatic operations such as counting, measuring, and localization. For example, as shown in Figure 2(1), when a task requires precise object counting, models that rely only on intrinsic perception often fail due to overlapping shapes, varying sizes, or background noise. In contrast, TVP enables the model to generate code that analyzes pixel-level cues such as color and boundary lines, allowing it to count objects more accurately. We evaluate this skill across three dimensions: **Attribute** (*i.e.*, color, shape, size), **Location** (*i.e.*, positions, distances, spatial relations), and **Symbolic** (*i.e.*, digits, letters, or graphical symbols).

**Abstraction**: This skill concerns the model's ability to transform low-level structured information into higher-level problem formulations. It requires not only recognizing surface patterns but also capturing the underlying rules and constraints, and converting them into computationally useful forms. For instance, as illustrated in Figure 2(6), the model must write code to abstract the puzzle into a network structure, representing each piece as a node and encoding adjacency relations as edges. This code-based abstraction allows the model to perform further search or optimization over the graph. In MMR-VIP, we evaluate abstraction across three dimensions: **Geometry** (*i.e.*, geometric formulations), **Physics** (*i.e.*, physical laws), and **Network** (*i.e.*, graph structures).

**Optimization**: This skill focuses on the model's ability to transform problem formulations into efficient algorithmic solutions. It requires not only identifying feasible solutions but also refining them to satisfy the given conditions. For example, as illustrated in Figure 2(7), the Rubik's Cube task requires the model to minimize the number of moves from an initial state to a target state. TVP enables the model to generate and execute code that systematically explores the space of valid cube operations, pruning redundant paths and converging to the optimal sequence of moves. We evaluate this skill across three dimensions: **Search** (*i.e.*, depth-first search, breadth-first search), **Planning** (*i.e.*, dynamic programming, linear programming), and **Computation** (*i.e.*, numerical calculations).

## 3.3 Benchmark Construction

To ensure that tasks are both solvable in the TVP paradigm and suitable for difficulty control, we adopt a Code2Task generation framework. We recruited five annotators with strong backgrounds in programming competitions and instructed them to write code that specifies task rules and automatically generates the corresponding images[2], problems, and answers. As task difficulty increased, annotators were required to design new rules and introduce greater computational complexity, thereby enriching the reasoning challenges. To facilitate this process, annotators were permitted to utilize

---

[2]We implemented visualization through `HTML` and `Matplotlib`.

Table 2: Experimental results on MMR-VIP. The best performance in each column is highlighted in **bold**. Red denotes cases where TVP underperforms CoT, while Green denotes cases where it outperforms CoT, with darker shades indicating larger magnitude of change.

| Model | | Difficulty Level | | | Cognitive Skill | | | | | | | | | Overall |
|---|---|---|---|---|---|---|---|---|---|---|---|---|---|---|
| | | Easy | Mid | Hard | Att | Loc | Sym | Geo | Phy | Net | Com | Sea | Pla | |
| **Open-source Models** | | | | | | | | | | | | | | |
| Keye-VL-1.5-8B | CoT | 28.0 | 11.4 | 4.8 | 12.0 | 9.3 | 21.2 | 11.2 | 8.3 | 27.5 | 35.0 | 15.6 | 14.3 | 14.8 |
| | T=1 | 9.1 | 3.2 | 2.5 | 4.2 | 3.8 | 5.2 | 3.8 | 4.4 | 12.5 | 6.7 | 4.2 | 3.8 | 4.9 (↓9.9) |
| Gemma-3-27B | CoT | 16.2 | 5.5 | 5.0 | 7.8 | 6.0 | 11.9 | 2.5 | 10.0 | 16.7 | 17.5 | 10.2 | 6.0 | 8.9 |
| | T=1 | 15.5 | 10.0 | 4.1 | 6.2 | 5.5 | 17.3 | 4.2 | 6.7 | 22.5 | 28.3 | 8.9 | 10.2 | 9.9 (↑1.0) |
| Qwen2.5-VL-7B | CoT | 13.2 | 7.0 | 3.9 | 7.5 | 3.6 | 7.3 | 3.8 | 8.9 | 26.7 | 6.7 | 6.8 | 6.4 | 8.0 |
| | T=1 | 7.3 | 5.5 | 1.6 | 2.5 | 2.9 | 7.5 | 2.1 | 2.2 | 17.5 | 21.7 | 5.3 | 1.7 | 4.8 (↓3.2) |
| Qwen2.5-VL-32B | CoT | 24.3 | 10.9 | 6.4 | 13.7 | 12.6 | 17.3 | 8.3 | 10.0 | 25.0 | 29.2 | 14.7 | 8.6 | 13.9 |
| | T=1 | 13.6 | 4.6 | 4.1 | 7.7 | 8.1 | 6.9 | 4.2 | 9.4 | 13.3 | 6.7 | 5.9 | 4.8 | 7.4 (↓6.5) |
| | T=3 | 18.9 | 9.8 | 6.8 | 10.0 | 8.8 | 18.3 | 2.5 | 7.2 | 20.8 | 28.3 | 13.3 | 10.7 | 11.8 (↓2.1) |
| Qwen2.5-VL-72B | CoT | 23.9 | 10.4 | 6.1 | 12.3 | 10.7 | 15.8 | 12.5 | 8.9 | 30.8 | 20.8 | 12.1 | 8.6 | 13.4 |
| | T=1 | 20.5 | 9.6 | 4.5 | 11.5 | 11.7 | 16.2 | 5.8 | 7.8 | 18.3 | 15.8 | 9.1 | 10.9 | 11.6 (↓1.8) |
| **Commercial Models** | | | | | | | | | | | | | | |
| GPT-4.1-mini | CoT | 42.7 | 20.2 | 9.8 | 23.3 | 23.8 | 32.1 | 19.6 | 21.1 | 30.8 | 34.2 | 18.9 | 26.9 | 24.2 |
| | T=1 | 45.5 | 28.2 | 14.1 | 16.0 | 23.3 | 49.4 | 22.1 | 22.2 | 35.0 | 39.2 | 24.7 | 37.1 | 29.3 (↑5.1) |
| | T=3 | 42.1 | 28.4 | 14.1 | 21.8 | 21.4 | 45.0 | 16.2 | 14.4 | 31.7 | 31.7 | 22.9 | 40.5 | 28.2 (↑4.0) |
| GPT-4.1 | CoT | 42.7 | 19.1 | 11.1 | 23.0 | 26.9 | 33.8 | 20.0 | 26.1 | 30.0 | 35.0 | 13.9 | 28.6 | 24.3 |
| | T=1 | 38.9 | 18.0 | 7.1 | 18.7 | 23.8 | 25.4 | 22.1 | 27.2 | 27.5 | 22.5 | 17.6 | 16.4 | 21.4 (↓2.9) |
| | T=3 | 47.1 | 25.5 | 12.1 | 18.7 | 25.0 | 50.4 | 13.8 | 32.8 | 28.3 | 32.5 | 20.3 | 36.9 | 28.3 (↑4.0) |
| Gemini-2.5-Flash | CoT | 46.4 | 18.0 | 10.9 | 17.8 | 27.6 | 42.5 | 26.2 | 25.6 | 32.5 | 40.0 | 17.7 | 28.1 | 25.1 |
| | T=1 | 32.7 | 14.5 | 7.9 | 9.0 | 17.4 | 34.2 | 22.1 | 25.0 | 12.5 | 32.5 | 14.4 | 19.5 | 18.3 (↓6.8) |
| | T=3 | 59.3 | **34.5** | 16.1 | 21.8 | 30.2 | **64.6** | 27.1 | **33.3** | 29.2 | 40.0 | **30.0** | 49.0 | 36.6 (↑11.5) |
| Gemini-2.5-Pro | CoT | 58.0 | 20.9 | 10.4 | 21.3 | 25.7 | 44.4 | 29.6 | 29.4 | 27.5 | 37.5 | 26.2 | 32.9 | 29.8 |
| | T=1 | 38.8 | 20.2 | 11.4 | 12.7 | 17.9 | 38.3 | 20.0 | 30.0 | 27.5 | 16.7 | 16.5 | 29.5 | 23.4 (↓6.4) |
| Claude-Sonnet-4 | CoT | 49.6 | 18.2 | 8.9 | 19.5 | 28.6 | 38.3 | 27.1 | 23.3 | 26.7 | 38.3 | 19.2 | 28.1 | 25.6 |
| | T=1 | 49.5 | 31.6 | 14.3 | 22.0 | 27.6 | 53.5 | 19.6 | 26.1 | 28.3 | **45.8** | 28.6 | 38.3 | 31.8 (↑6.2) |
| **Native TVP Models** | | | | | | | | | | | | | | |
| o4-mini | | 57.7 | 30.7 | **17.3** | 28.0 | 24.8 | 55.4 | 25.4 | 26.1 | 35.0 | 40.0 | 27.3 | 55.2 | 35.2 |
| GPT-5-mini | | 61.8 | 29.8 | 16.8 | **32.7** | 29.8 | 52.9 | 27.9 | 26.1 | 32.5 | 42.5 | 28.3 | 53.3 | 36.1 |
| GPT-5 | | **65.5** | 33.8 | 15.4 | 27.5 | **31.7** | 60.4 | **30.8** | 31.1 | **39.2** | 39.2 | 28.2 | **56.9** | **38.2** |
| **Reference** | | | | | | | | | | | | | | |
| Human | | 69.6 | 55.4 | 35.7 | 48.3 | 54.8 | 75.0 | 37.5 | 27.8 | 58.3 | 66.7 | 50.0 | 69.1 | 53.6 |

AI-assisted code editors (*e.g.*, Cursor). Finally, we conducted cross-validation of all generated code to verify correctness, where each program was independently reviewed by multiple annotators.

In total, MMR-VIP encompasses **28** carefully crafted task types, each designed across three difficulty levels. For every task and difficulty, we randomly generated **20** instances, resulting in a benchmark of **1,680** instances in total. We include detailed examples of each task in the Appendix C. The mapping between task types and their corresponding cognitive skills is presented in Table 1. Since all tasks are synthesized from code, MMR-VIP is reproducible and extendable. Researchers can regenerate new instances by adjusting parameters or extend the benchmark with new task rules, making MMR-VIP a continuously evolvable framework rather than a fixed dataset.

## 4 EXPERIMENTS

In this section, we present a comprehensive evaluation of existing MLLMs on MMR-VIP. We systematically evaluate model performance across different difficulty levels and cognitive skills, and further contrast the effectiveness of CoT and TVP. We also analyze from multiple perspectives, including the effect of iteration rounds, the role of input modalities, and the distribution of error types.

### 4.1 EXPERIMENTAL SETUP

We evaluate three categories of MLLMs on MMR-VIP: commercial models (*e.g.*, GPT-4.1 (OpenAI, 2025b), Gemini-2.5-Flash, Gemini-2.5-Pro (DeepMind, 2025), Claude-Sonnet-4 (Anthropic, 2025)), open-source models (*e.g.*, Qwen2.5-VL (Bai et al., 2025), Gemma-3 (Kamath et al., 2025), Keye-VL-1.5 (Yang et al., 2025a)), and native TVP models (*e.g.*, o4-mini, GPT-5). We do not include existing open-source models designed specifically for Thinking with Images, since these models primarily focus on applying fixed transformations to images rather than freely generating

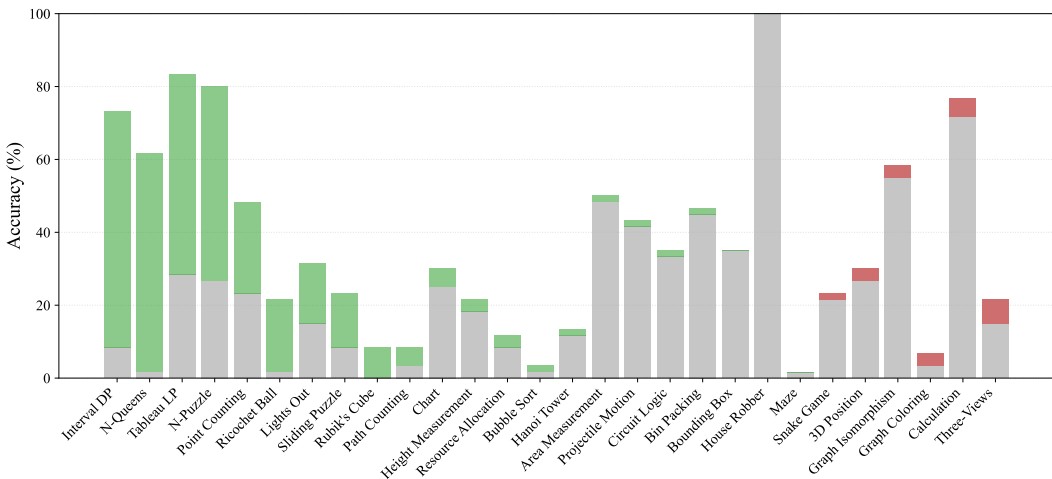

Figure 3: Performance comparison of Gemini-2.5-Flash on different tasks under CoT and TVP ($T = 3$). Gray indicates the baseline performance of CoT, Green indicates improvements of TVP over CoT, and Red indicates degradations of TVP over CoT.

code to support reasoning. Moreover, to assess the effectiveness of different reasoning strategies, we compare three settings: **Chain-of-Thought**, **single-turn TVP**, where the model invokes the code executor once, and **multi-turn TVP**, where the model can iteratively generate, execute, and refine code for up to $T = 3, 5, 7$ rounds. We provide the detailed prompts used for all settings in the Appendix D. As a reference, we randomly sample 168 instances and invite human participants to solve these tasks. Each participant is allowed to leverage search engines and interpreters during the process. We adopt accuracy as the evaluation metric. We report results along three perspectives: performance across different difficulty levels, performance across distinct cognitive skills, and the overall accuracy.

## 4.2 EXPERIMENTAL RESULTS

As shown in the Table 2, our experiments on MMR-VIP yield several key findings:

(1) **Performance differences across model types and reasoning paradigms.** For open-source models like Qwen2.5-VL-72B, introducing TVP offers negligible gains and sometimes results in performance drops, owing to their limited visual programming capabilities. For commercial models, single-turn code execution produces unstable results, whereas multi-turn execution consistently yields significant improvements. For instance, Gemini-2.5-Flash shows an accuracy gain of 18.3% when increasing from $T = 1$ to $T = 3$. For native TVP models, although GPT-5 achieves the highest performance, it attains only **38.2%** accuracy, reflecting the substantial limitations that remain. We can observe a clear performance gap relative to humans, underscoring that humans are more adept at leveraging external tools to solve complex visual problems.

(2) **Clear difficulty ladder.** The results align closely with the benchmark's design, showing a distinct ladder-shaped performance trend. Compared to direct CoT, TVP shows negligible differences on easy tasks, achieves the largest improvements on medium tasks, and delivers consistent gains on hard tasks. Nevertheless, performance at the hard level remains very low, with the best accuracy reaching only 17.3%. This demonstrates that MMR-VIP effectively stratifies problems by difficulty, thereby exposing the limits of current MLLMs' reasoning capabilities.

(3) **Imbalanced cognitive skills.** The results reveal marked disparities across cognitive skills. TVP delivers the most significant improvements in Optimization, where models effectively leverage programmatic search, planning, and computation to tackle complex problem-solving tasks. As shown in Figure 3, Gemini-2.5-Flash exhibits large gains on tasks such as *Interval Dynamic Programming*, *N-Queens*, *Tableau Linear Programming*, and *N-Puzzle*, where code execution is essential to explore solution spaces. In addition, TVP also enhances Symbolic perception, since code allows models to precisely recognize, parse, and manipulate digits, letters, or graphical symbols. However, performance in Abstraction remains the most challenging, where models still struggle to translate

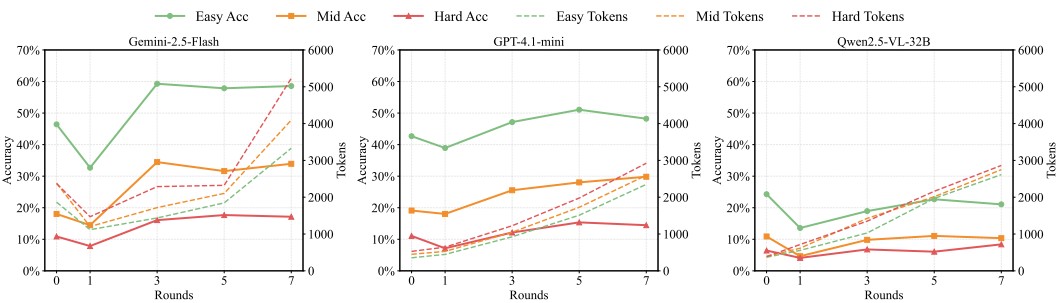

Figure 4: Performance and token consumption under different iteration rounds ($T = 0, 1, 3, 5, 7$). $T = 0$ corresponds to CoT. Green, orange, and red correspond to Easy, Medium, and Hard levels.

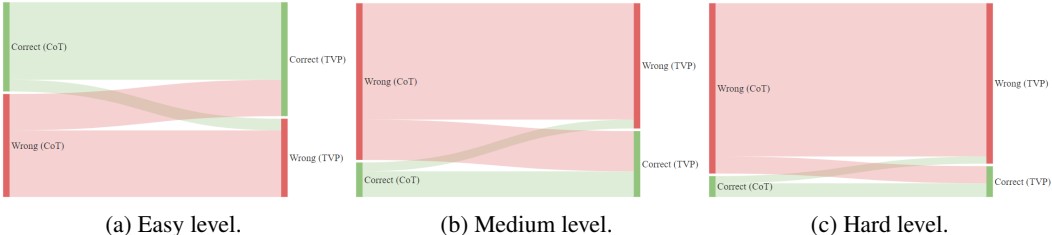

(a) Easy level.  (b) Medium level.  (c) Hard level.

Figure 5: Correctness flow between CoT ($T = 0$) and TVP ($T = 3$) for Gemini-2.5-Flash.

low-level visual cues into high-level formulations such as geometric equations, physical laws, or graph structures. This underscores the necessity of improving their ability to abstract through code.

### 4.3 ANALYSIS

#### 4.3.1 IMPACT OF ITERATION ROUNDS

We examine the impact of iterative rounds of code execution on model performance across easy, medium, and hard tasks. As shown in Figure 4, compared to direct CoT, single-turn TVP ($T = 1$) often leads to a drop in accuracy. To better understand this phenomenon, we compute the correlation between the performance difference of TVP ($T = 1$) versus CoT and the success rate of program execution. The Pearson correlation coefficient is **0.81** ($p \approx 0.05$), indicating a strong positive relationship. A primary source of degradation arises when incorrect code execution propagates interpreter error messages into the reasoning process, thereby misguiding subsequent inference.

Performance generally peaks at $T = 3$ or $T = 5$, where iterative refinement enables more reliable program execution and reflective reasoning. As illustrated in Figure 5, we further analyze the correctness flow between CoT and TVP ($T = 3$). The results show that the most significant changes occur at the Medium difficulty level. However, for open-source models like Qwen2.5-VL-32B, additional iterations fail to bring noticeable gains. This finding highlights that robust visual programming capabilities are indispensable for open-source models to fully realize the benefits of TVP. Meanwhile, increasing to $T = 7$ brings little to no additional gains and instead results in significantly higher token consumption, highlighting the trade-off between accuracy and efficiency.

#### 4.3.2 INFLUENCE OF INPUT MODALITIES

To further investigate the role of input modalities in TVP, we select four tasks from MMR-VIP that can be represented in both textual and visual forms: *Tableau LP*, *Chart*, *Graph Coloring*, and *Maze*. This design allows us to directly compare model performance under three conditions: (1) image-only input (I), (2) text-only input (T), and (3) combined image–text input (I & T). Results in Table 3 show that text input generally outperforms image input, indicating that current models still have weaker visual reasoning capabilities. Moreover, visual inputs sometimes introduce perception errors, which can

Table 3: Performance comparison under different input modalities.

| Model | I | T | I & T |
|---|---|---|---|
| GPT-4.1-mini | 26.3 | 75.0 | 76.3 |
| Claude-Sonnet-4 | 7.5 | 50.0 | 63.8 |
| GPT-5-mini | 5.0 | 50.0 | 53.8 |
| GPT-5 | 25.0 | 46.3 | 70.0 |

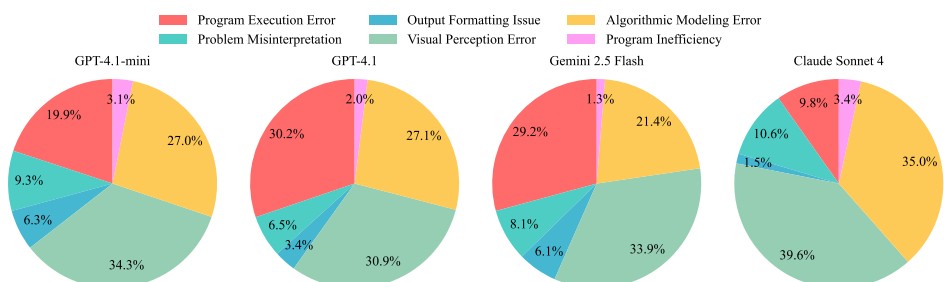

Figure 6: Error analysis of four models under TVP ($T = 1$).

propagate through subsequent reasoning steps. Nevertheless, combined multimodal input consistently surpasses unimodal input, particularly on tasks where the visual layout conveys structural or spatial constraints that are difficult to capture with text alone.

### 4.3.3 ERROR ANALYSIS

To better understand the limitations of TVP, we conduct a detailed error analysis by categorizing incorrect predictions into six major types: Program Execution Error, Visual Perception Error, Algorithmic Modeling Error, Program Inefficiency, Problem Misinterpretation, and Output Formatting Issue. The precise definitions and representative examples of each category are provided in the Appendix F. As illustrated in Figure 6, the most common sources of error are Visual Perception Error, Algorithmic Modeling Error, and Program Execution Error. These results align with our earlier findings: they reflect (1) the insufficiency of models in Perception and Abstraction, where they struggle to accurately extract information from visual inputs and transform it into computationally useful formulations, and (2) the limitations of current models' programming capabilities, where code errors remain prevalent. We also provide several case studies of CoT and TVP in Appendix G.

## 5 RELATED WORKS

**Multimodal Reasoning.** Multimodal reasoning has recently become a prominent frontier in AI research, with an expanding set of benchmarks and investigations underscoring its pivotal importance across domains such as interpreting scientific diagrams (Yue et al., 2024; Guo et al., 2025), solving geometry problems (Zhang et al., 2024b; Wang et al., 2024), and tackling visual puzzles (Chia et al., 2024; Ghosal et al., 2025; Song et al., 2025). Recent work (Huang et al., 2025; Meng et al., 2025; Chris et al., 2025; Hong et al., 2025; Deng et al., 2025; Wang et al., 2025c;b) has focused on enhancing models' reasoning ability through reinforcement learning, thereby extending reasoning depth, enabling reflection and verification, and improving performance on complex tasks. However, some studies argue that RL is constrained by an invisible leash (Wu et al., 2025a), preventing it from discovering new reasoning trajectories beyond the model's initial capabilities (Lin & Xu, 2025).

**Visual Programming.** Visual programming (Yang et al., 2025b; Surís et al., 2023; Hu et al., 2024b) requires models to generate executable code based on visual inputs. MMCode (Li et al., 2024) evaluates MLLMs' code generation abilities on competitive-programming problems presented with visual contexts. HumanEval-V (Zhang et al., 2024a) is a benchmark designed to evaluate complex diagram understanding and visual reasoning abilities in programming contexts. It assesses whether models can accurately infer the underlying rules embedded in visual diagrams and subsequently generate correct programs that satisfy the corresponding test cases. Moreover, SWE-bench Multimodal (Yang et al., 2025b) evaluates agents on their ability to fix bugs in visual, user-facing JavaScript software, with tasks that incorporate images within their problem statements or test cases. Built upon the Mini-level of the XLogoOnline platform, XLogoOnline-Mini (Wen et al., 2025) requires models to synthesize programs that control a turtle navigating through a grid to accomplish a specified goal. The benchmark evaluates a broad spectrum of capabilities, including mathematical reasoning, logical reasoning, spatial understanding, and planning. The primary difference of our work, MMR-VIP, is that it aims to evaluate a model's multimodal reasoning capabilities, where code serves only as an optional tool to enhance reasoning rather than being the final output. All code generated in MMR-VIP is free-form and intended solely to assist in problem-solving.

**Tool-Integrated Reasoning.** Rather than relying solely on parametric knowledge within the model, tool-integrated reasoning (TIR) (Jin et al., 2025; Li et al., 2025; Xue et al., 2025; Feng et al., 2025; Dong et al., 2025) enables the model to reason with external tools, such as a Python interpreter. Extending this idea to multimodal settings, the paradigm of Thinking with Images (TWI) has emerged as an effective approach (Lu et al., 2025; Su et al., 2025b;a; Lai et al., 2025; Wang et al., 2025d; Wu et al., 2025b; Zhou et al., 2025). Instead of relying solely on textual reasoning, models are equipped with a predefined set of visual tools such as cropping, zooming, or rotating, which allow them to refine perception during problem solving. Recently, there has been a growing trend of enabling MLLMs to generate executable code as part of the reasoning process (Tang et al., 2025; Zhao et al., 2025; Hu et al., 2024a; Zhang et al., 2025; Wang et al., 2025a), showcasing the potential of the TVP.

## 6    CONCLUSION

In this work, we introduced MMR-VIP, a benchmark designed to evaluate multimodal agentic reasoning under the Thinking with Visual Programming paradigm. Beyond text-based CoT and fixed visual tools, TVP allows models to flexibly generate, execute, and refine programmatic code, which serve as intermediate reasoning steps to facilitate multimodal problem solving. MMR-VIP is specifically crafted for this paradigm, featuring problems that are unsolvable under CoT-based reasoning but become tractable when integrated with visual programming interactions. Progress in multimodal agentic reasoning will depend critically on strengthening models' coding proficiency, enhancing their visual abstraction ability, and equipping them with multi-round iterative reasoning strategies.

ETHICS STATEMENT

All experimental procedures involving human participants were conducted in accordance with the relevant ethical guidelines. Moreover, all data instances in our benchmark are puzzle-style problems that are automatically synthesized through scripts rather than collected from real-world human data. As such, the dataset contains no personal, harmful, or biased information. This ensures that MMR-VIP is entirely safe for research and avoids introducing any sensitive or ethically problematic content.

REPRODUCIBILITY STATEMENT

Our dataset is entirely script-synthesized rather than manually annotated or generated by LLMs, ensuring full reproducibility. To facilitate this, we will release the synthesis scripts with fixed random seeds alongside the final MMR-VIP dataset. We also provide data examples in the supplementary materials. In addition, we provide detailed prompts used in all experiments in Appendix D, and we will open-source the evaluation code together with the Python interpreter environment. This guarantees that researchers can faithfully reproduce our experimental results.

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

## A   LLM USAGE STATEMENT

In this work, Large Language Models (LLMs) were used solely as general-purpose auxiliary tools. Their role was limited to polishing grammar and phrasing to enhance the clarity of the manuscript, as well as assisting in the generation of Python and LaTeX code for creating figures and tables. No parts of the research ideation, experimental design, analysis, or substantive writing relied on LLMs.

## B   DISCUSSION

Here, we would like to discuss the relationship between Thinking with Images (TWI) and Thinking with Visual Programming (TVP).

Existing approaches under the Thinking with Images paradigm typically rely on a predefined set of visual tools, such as cropping, zooming, and rotating. These operations can indeed enhance perceptual accuracy, especially for handling high-resolution images or focusing attention on relevant regions. However, their scope is inherently narrow. While effective for improving low-level perception, such fixed transformations provide limited support for deep reasoning tasks that require abstraction, planning, or algorithmic optimization. In other words, current Thinking with Images primarily enhances *seeing more carefully*, but does not necessarily enable *thinking more deeply*.

In contrast, Thinking with Visual Programming generalizes beyond fixed toolkits by allowing models to write and execute code, thus treating visual operations themselves as programmable functions. This enables not only flexible tool selection but also the creation of new tools on demand, allowing the reasoning process to adapt dynamically to the task at hand. Under this view, cropping or rotating an image represents only one instance within a broader spectrum of programmable operations, which may also involve algorithmic simulation, complex computation, or visualization.

From this perspective, TWI can be regarded as a subset of TVP, serving as a valuable stepping stone but not the ultimate goal. As our experimental results demonstrate, current models remain far from fully realizing the TVP paradigm. While existing studies have already achieved promising outcomes under the TWI framework, a substantial gap persists between these methods and the broader vision of TVP. Bridging this gap requires equipping models with stronger visual programming capabilities and more advanced visual abstraction skills, enabling them to move beyond fixed perceptual tools toward flexible, programmable reasoning. On this foundation, agentic reinforcement learning can become truly effective. In the future, we envision equipping MLLMs with access to external resources such as web browsers. This would allow them not only to autonomously create tools through code but also to search for and integrate existing tools from the internet.

## C  BENCHMARK DETAILS

To ensure the data quality of MMR-VIP, we provided annotators with a detailed guideline:

All tasks must:
(1) Be code-synthesizable (problems, images, and solutions are generated by code).
(2) Be aligned with cognitive skills (at least 1, at most 3 from the given taxonomy).
(3) Be stratified into difficulty levels (Easy / Medium / Hard).
(4) Be suitable for programmatic reasoning (problems solvable or aided by code execution).
**Cognitive Skills**
(1) Attribute: identify colors, shapes, sizes.
(2) Location: detect positions, distances, spatial relations.
(3) Symbolic: recognize digits, letters, or visual symbols.
(4) Geometry: formulate geometric equations or relations.
(5) Physics: model dynamics using physical laws.
(6) Network: construct graph structures (nodes, edges, constraints).
(7) Search: implement DFS, BFS, or other exploration methods.
(8) Planning: apply dynamic/linear programming to solve constrained problems.
(9) Computation: perform numerical calculations or algorithmic procedures.
**Difficulty Levels**
(1) Easy: solvable using intrinsic perceptual and reasoning abilities, without code execution.
(2) Medium: requiring programmatic operations, where external computation is essential.
(3) Hard: remaining challenging even with programming support, typically due to high algorithmic complexity or intricate constraints.
**Workflow**
Annotators should first define the problem (including its target cognitive skills and difficulty levels), then implement code that generates instances and computes the ground-truth solution. Next, the problem must be visualized using standard libraries to ensure clarity. Each program should support batch generation of images, questions, and answers across difficulty levels. Finally, the generated code must undergo validation, where outputs are independently reviewed to ensure correctness and consistency between problem, visualization, and answer.

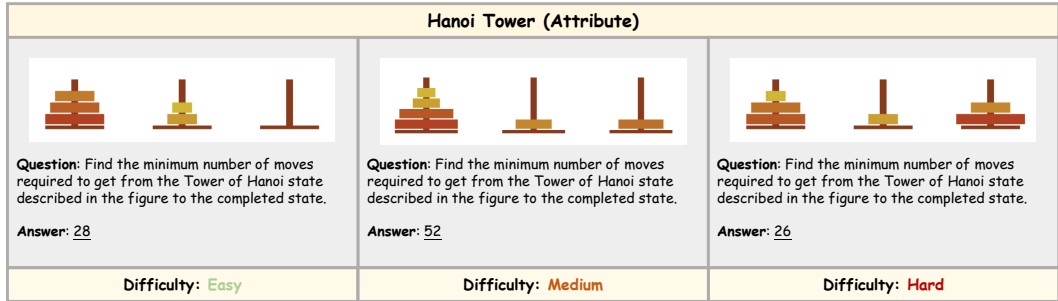

Figure 7: Data example of Hanoi Tower.

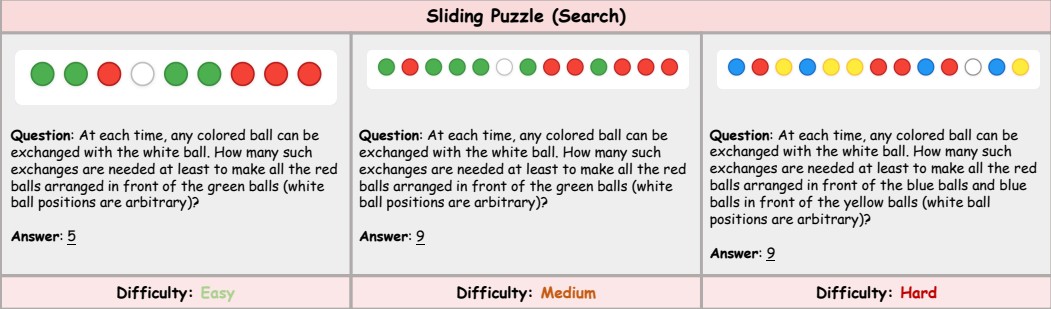

Figure 8: Data example of Sliding Puzzle.

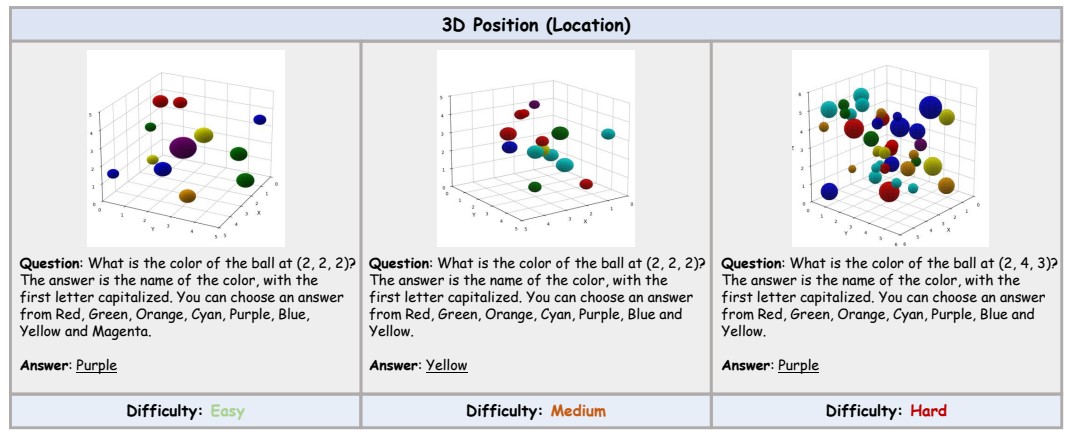

Figure 9: Data example of 3D Position.

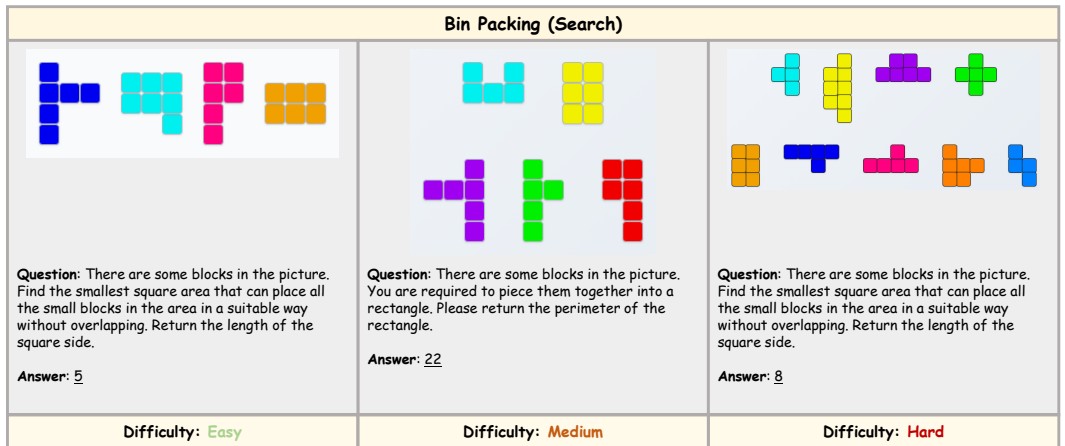

Figure 10: Data example of Bin Packing.

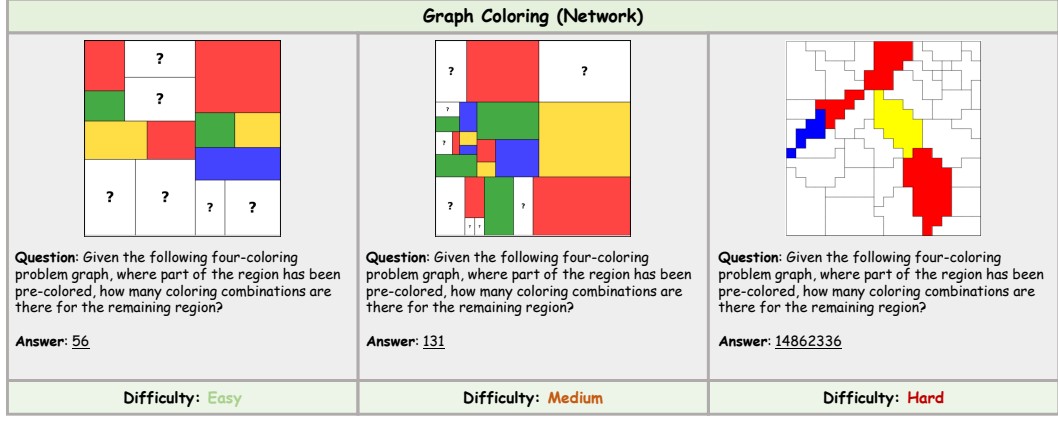

Figure 11: Data example of Graph Coloring.

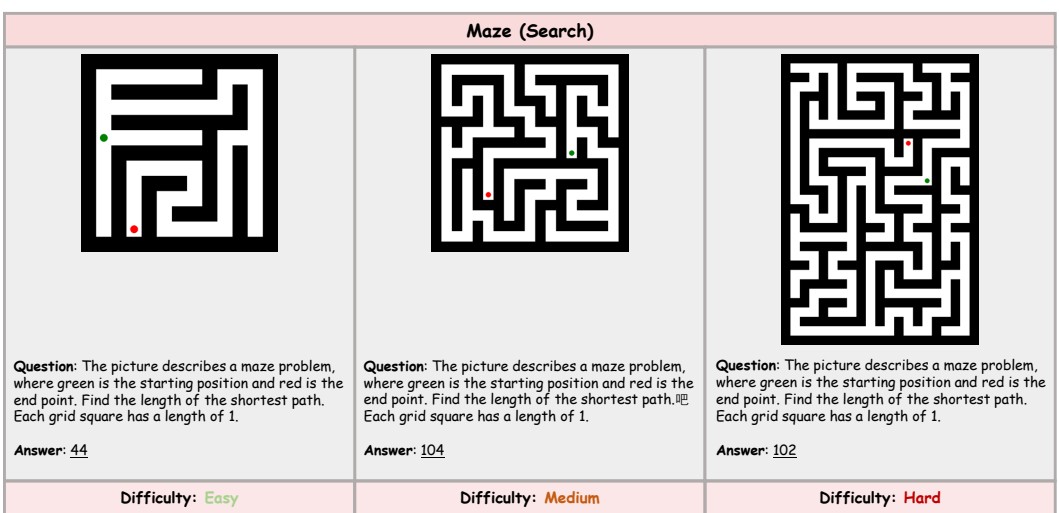

Figure 12: Data example of Maze.

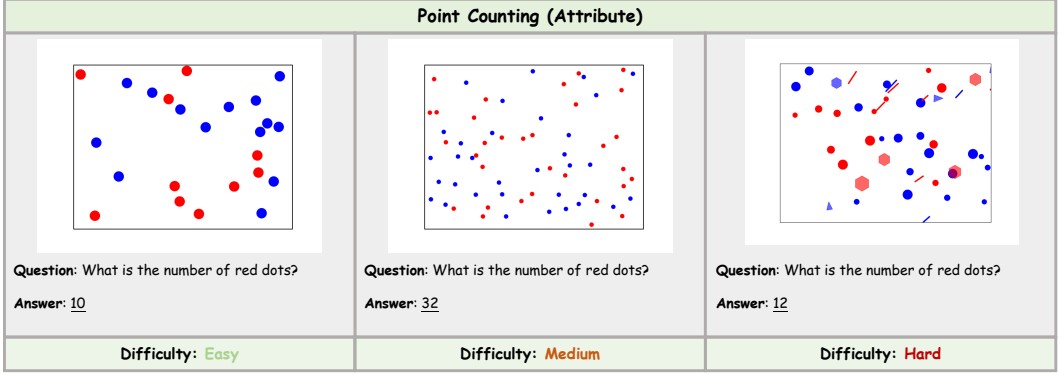

Figure 13: Data example of Point Counting.

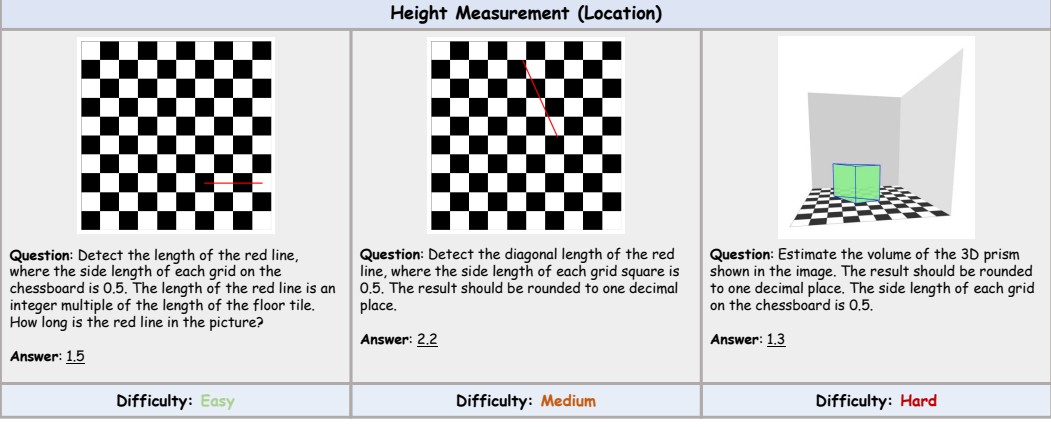

Figure 14: Data example of Height Measurement.

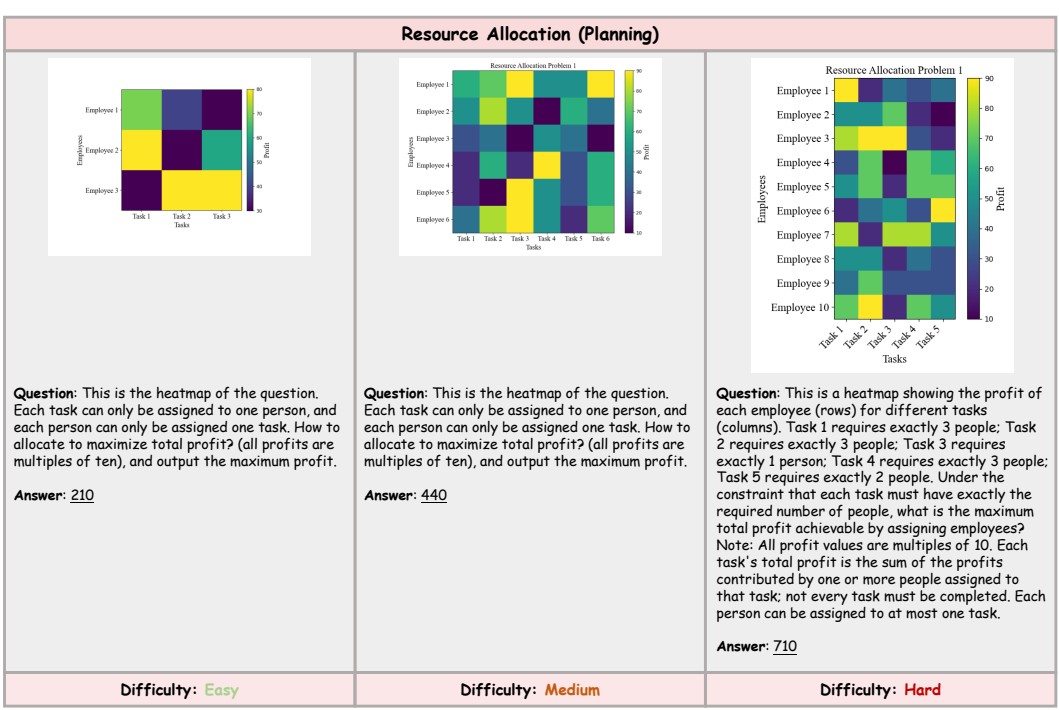

Figure 15: Data example of Resource Allocation.

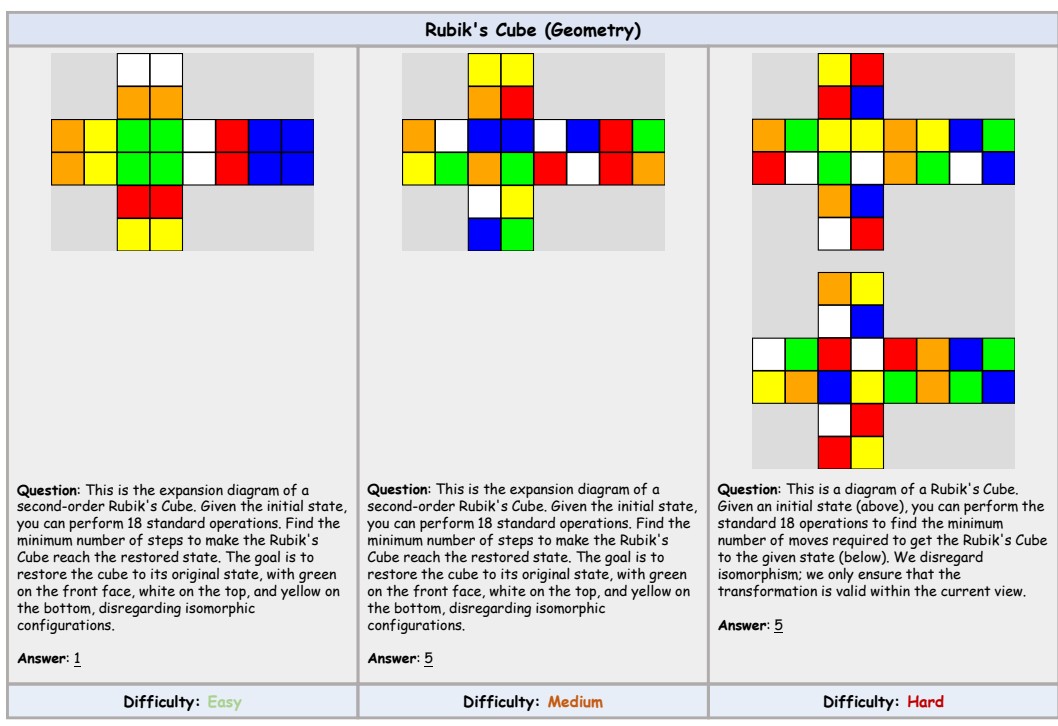

Figure 16: Data example of Rubik's Cube.

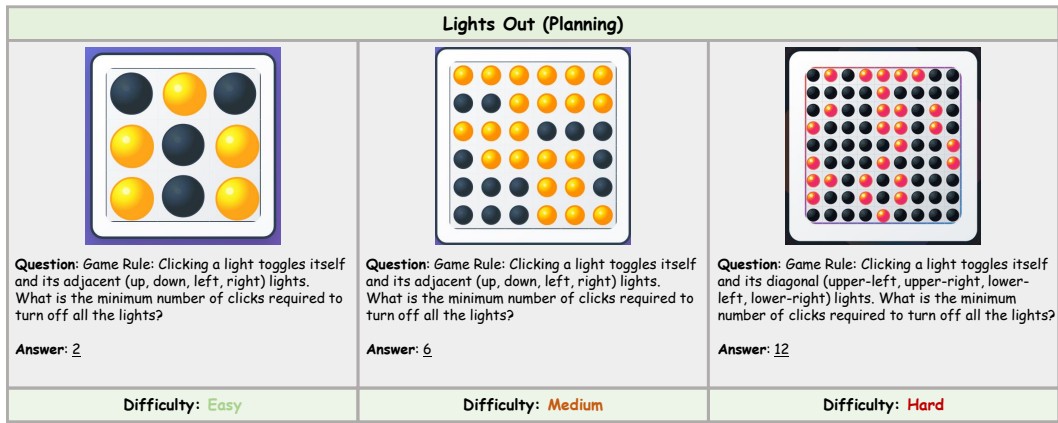

Figure 17: Data example of Lights Out.

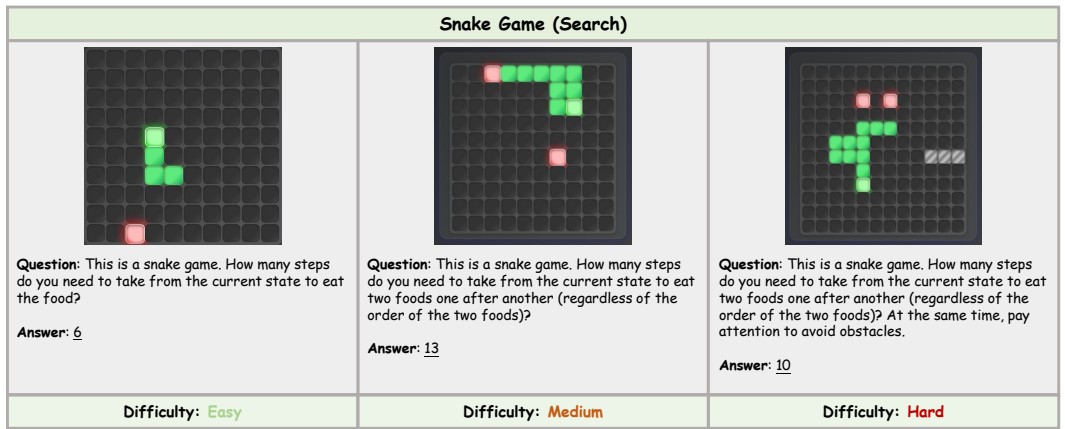

Figure 18: Data example of Snake Game.

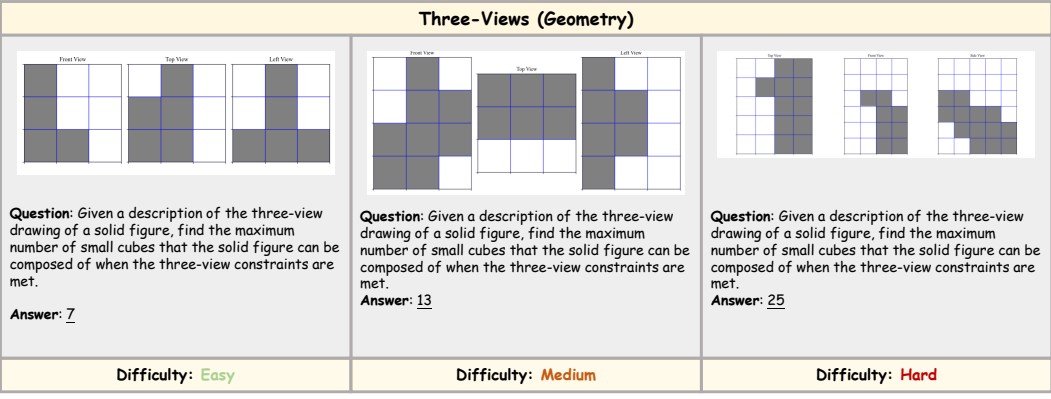

Figure 19: Data example of Three-Views.

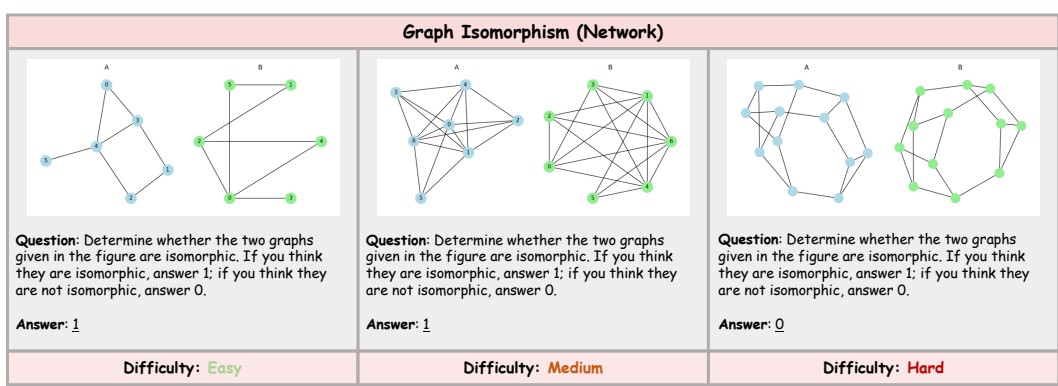

Figure 20: Data example of Graph Isomorphism.

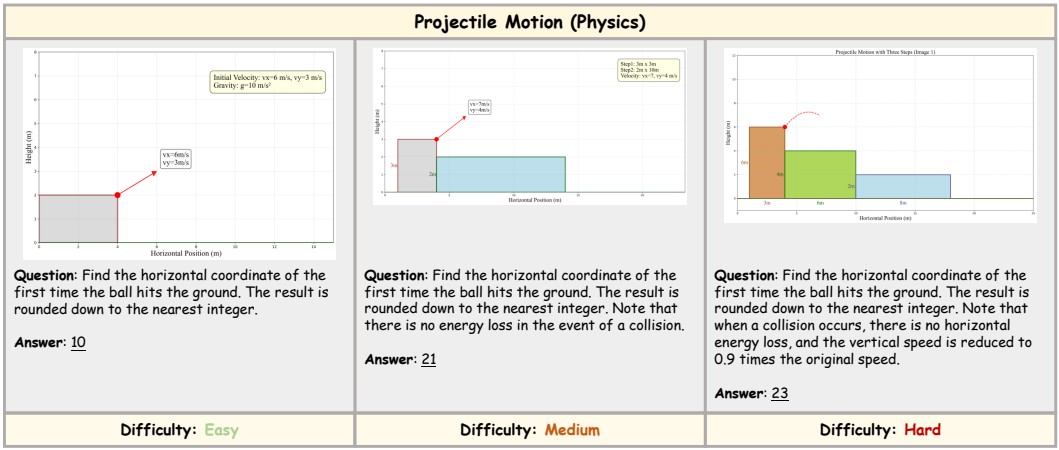

Figure 21: Data example of Projectile Motion.

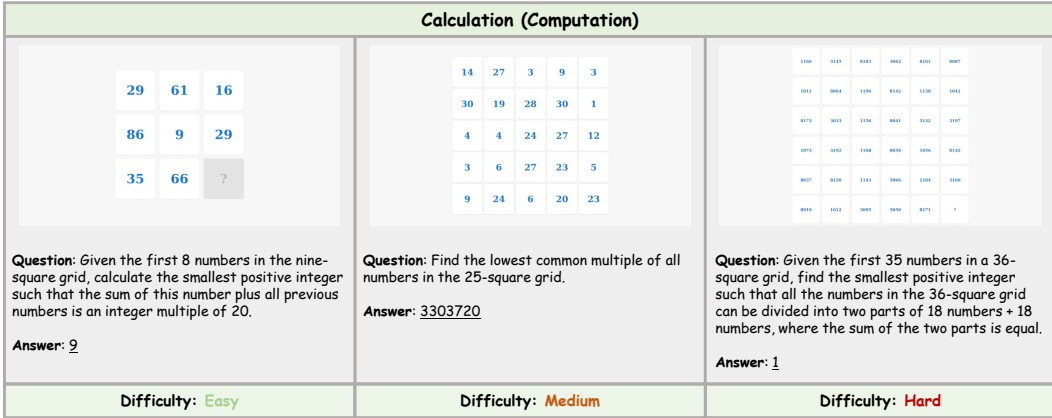

Figure 22: Data example of Calculation.

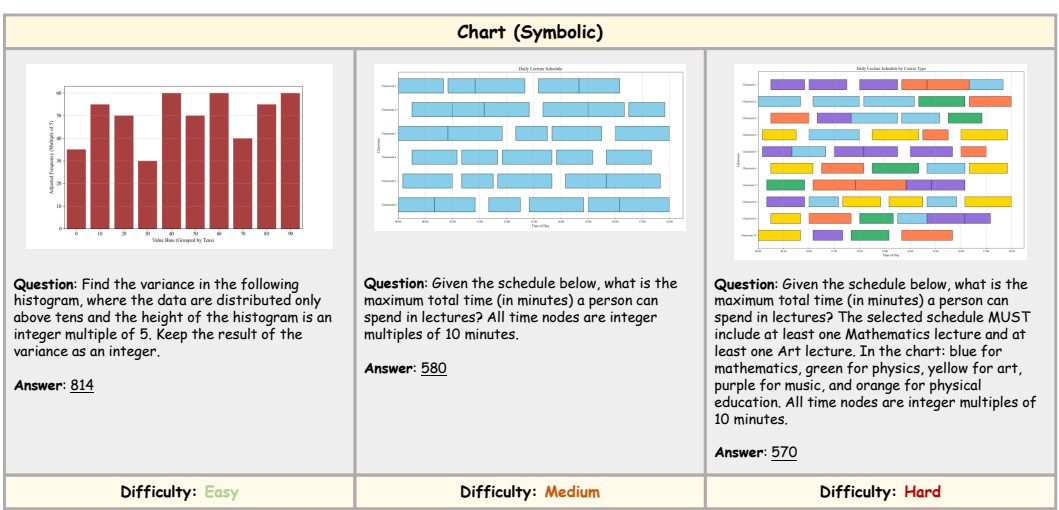

Figure 23: Data example of Chart.

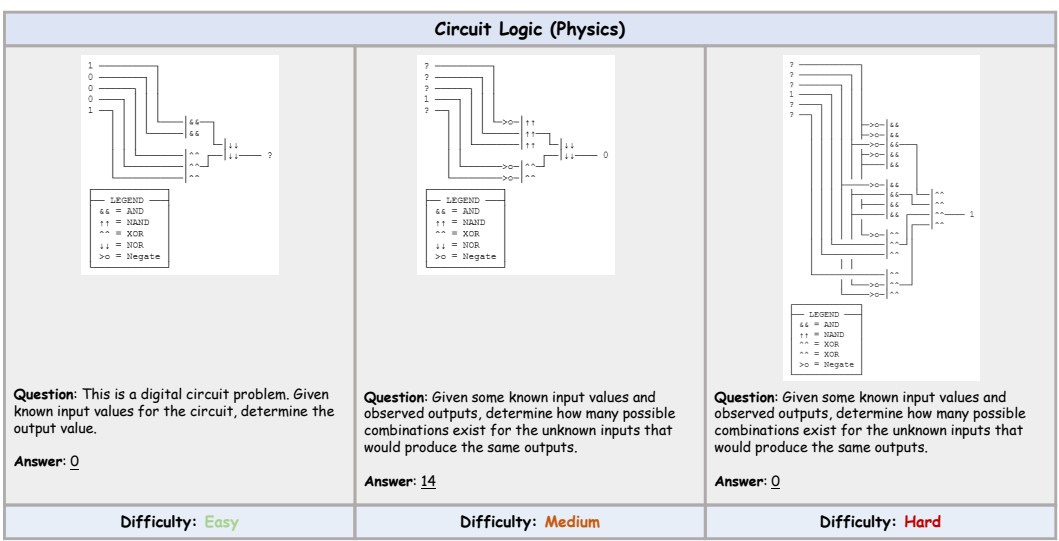

Figure 24: Data example of Circuit Logic.

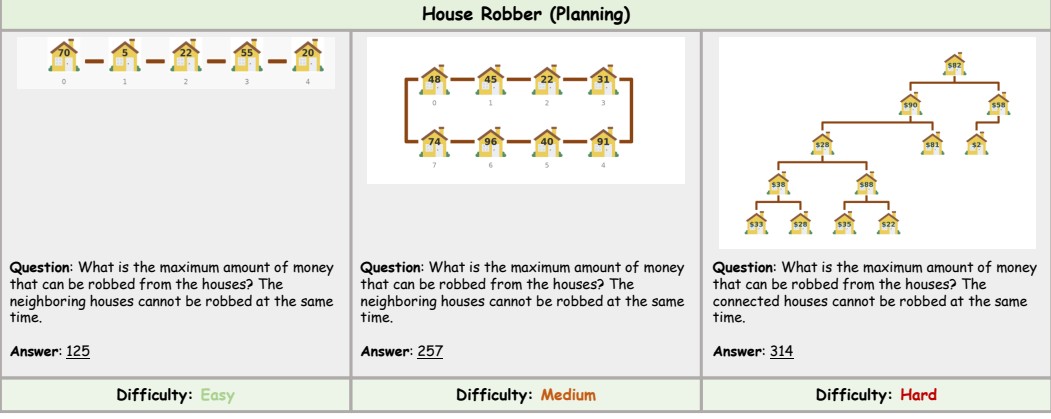

Figure 25: Data example of House Robber.

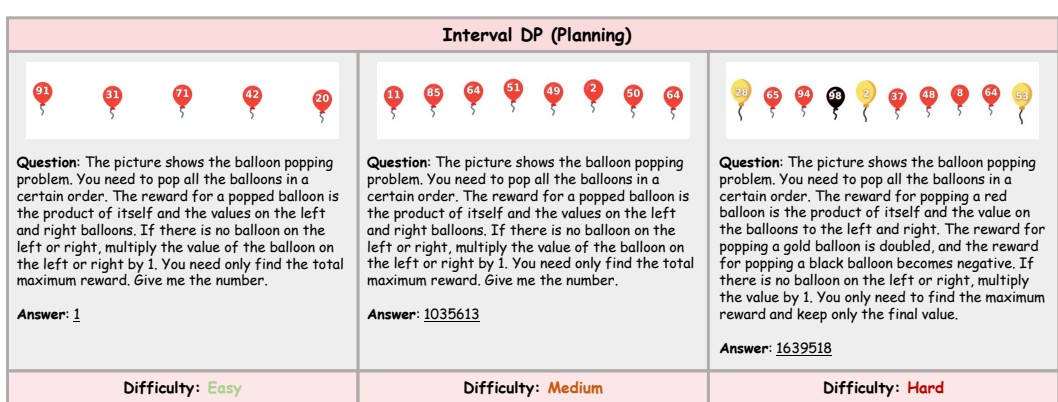

Figure 26: Data example of Interval DP.

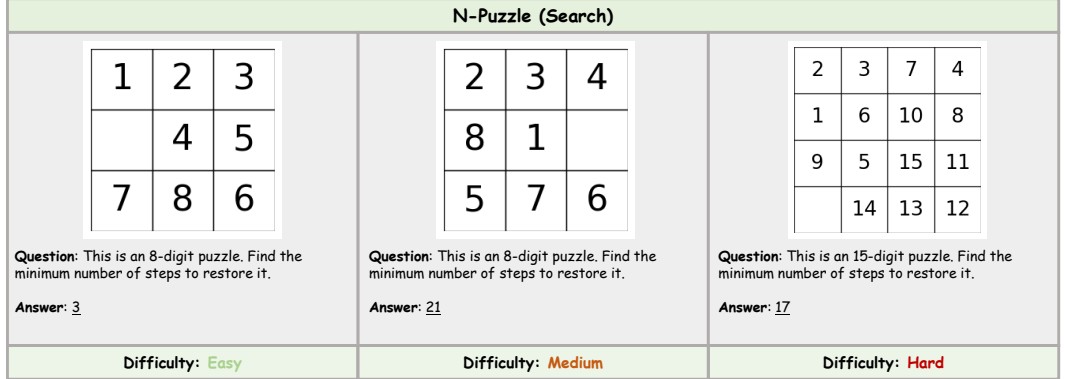

Figure 27: Data example of N-Puzzle.

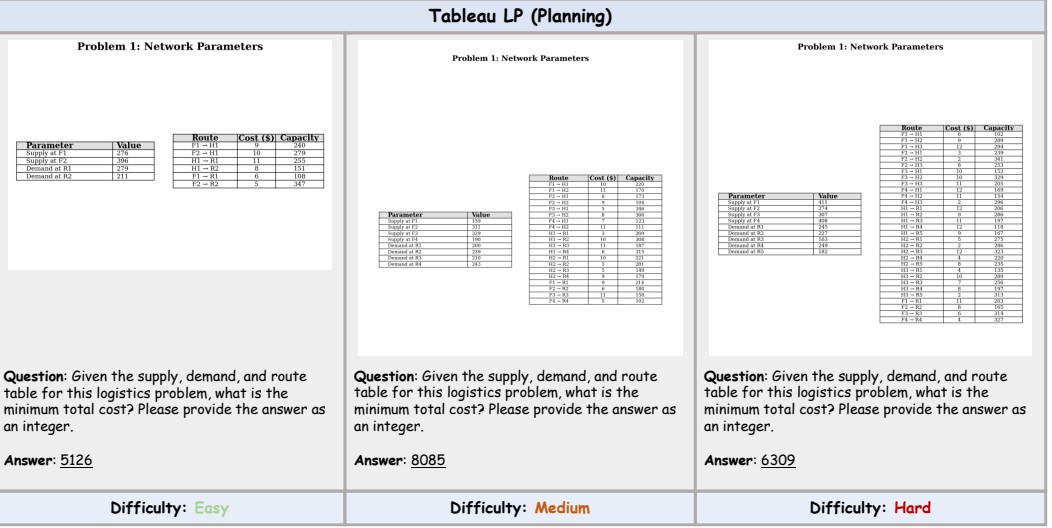

Figure 28: Data example of Tableau LP.

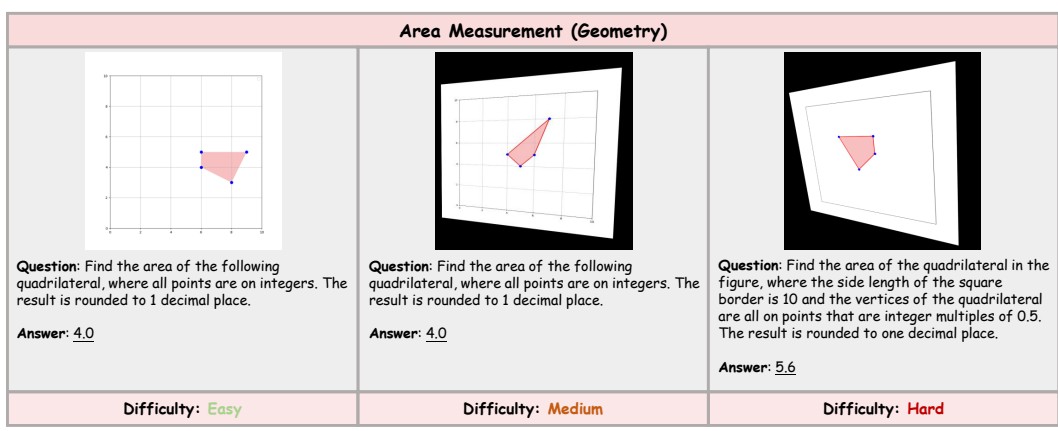

Figure 29: Data example of Area Measurement.

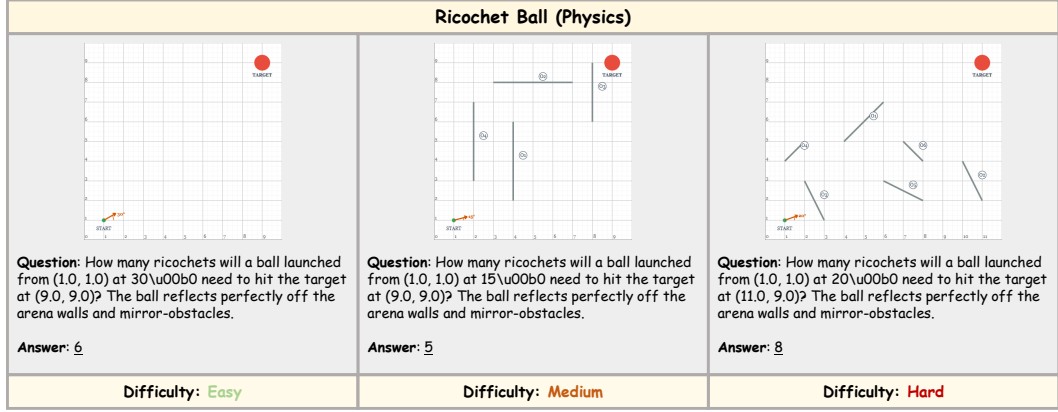

Figure 30: Data example of Ricochet Ball.

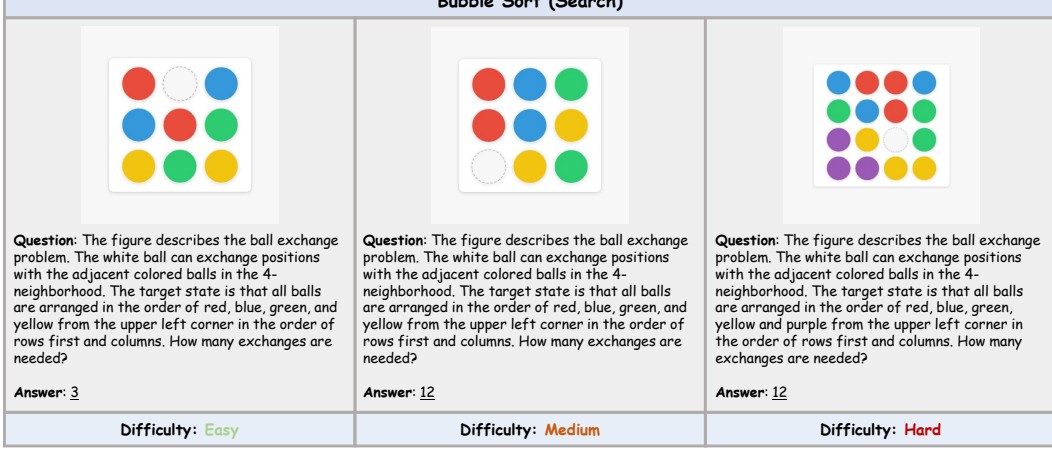

Figure 31: Data example of Bubble Sort.

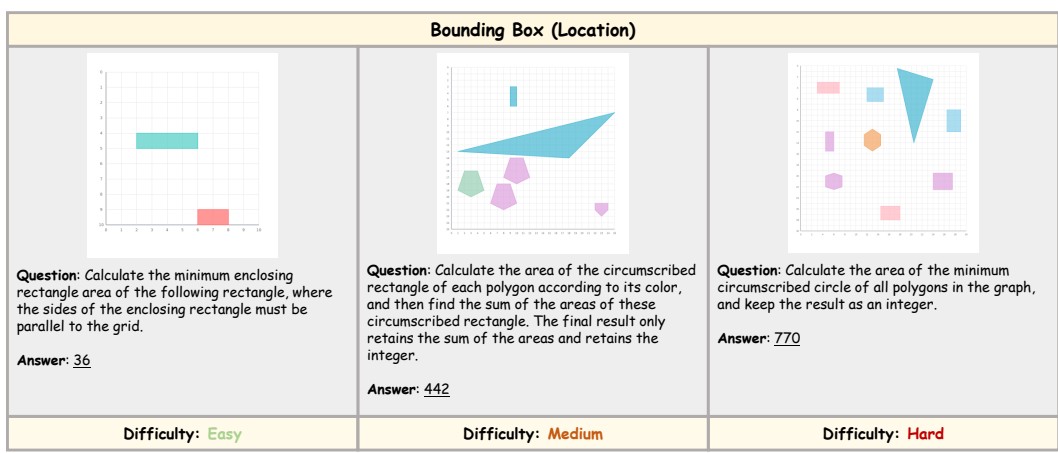

Figure 32: Data example of Bounding Box.

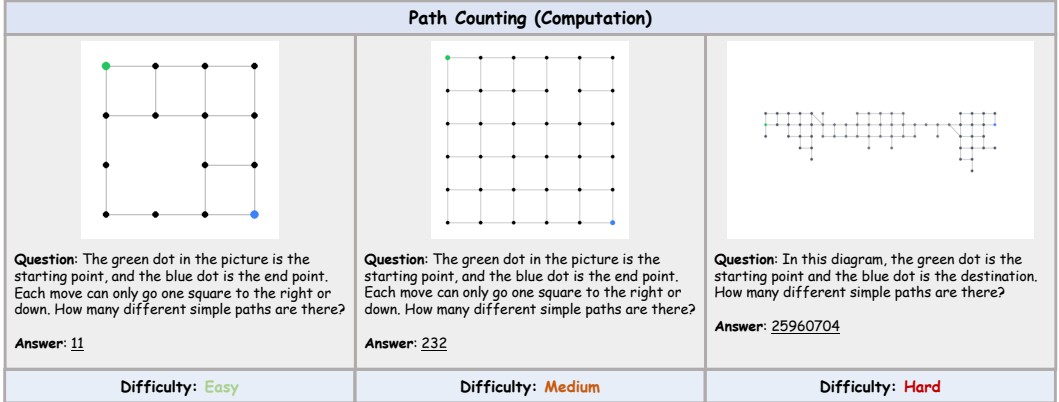

Figure 33: Data example of Path Counting.

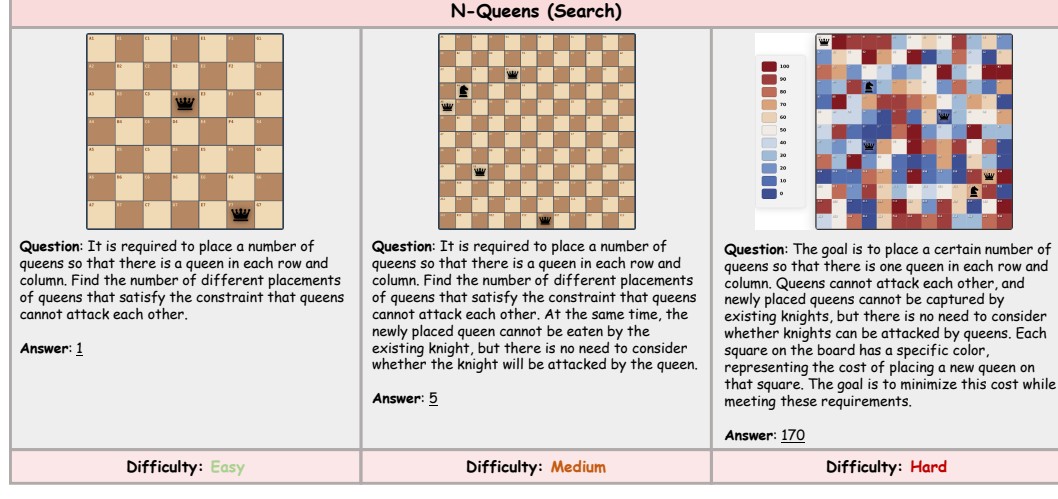

Figure 34: Data example of N-Queens.

# D EVALUATION DETAILS

We provide the prompts for both direct CoT reasoning and multi-turn TVP reasoning, as illustrated in Figure 35 and Figure 36.

---

**Prompt for Direct CoT**

---

**System Prompt:**
You FIRST think about the reasoning process as an internal monologue and then provide the final answer. The reasoning process MUST BE enclosed within <think> </think> tags. The final answer MUST BE put in \\boxed{}. Please note that if the answer requires a numerical value, please keep only the number without punctuation, units, formulas or explanations. Don't run code in your own environment.

---

Figure 35: Prompt for direct CoT reasoning.

---

**Prompt for Multi-turn TVP**

---

**System Prompt:**
You are a visual reasoning assistant that MUST write executable Python code to solve problems. You can iterate through multiple rounds to refine your solution (maximum {N} code executions).
IMPORTANT CODE FORMATTING RULES:
- You MUST wrap your code EXACTLY with  and  tags
- Do NOT use backticks (`), triple-backticks (```), or any other delimiters
- Inside ... put only valid Python code
- Do NOT HTML-escape characters (use <, >, &, not <, >, &)
HELPER FUNCTIONS:

```python
import os
import re
import typing
def find_original_image_name(work_dir: str = '.') -> typing.Optional[str]:
    '''Find the original image filename, excluding processed versions'''
    for f in sorted(os.listdir(work_dir)):
        if not f.lower().endswith('.png'): continue
        if f.startswith('crop_'): continue
        if re.search(r'_m(?:\\d+)?\\.png$', f): continue
        return f
    return None
def processed_image_name(original_image: str) -> str:
    '''Return processed image filename for current iteration'''
    base, ext = os.path.splitext(original_image)
    return f'{{base}}_m{iteration}{{ext}}'
```

CODE REQUIREMENTS:
- Use `find_original_image_name()` to locate the input image
- Save your processed image using `processed_image_name()` (will be *_m{iteration}.png)
- Use only relative paths and work within the current directory
- Do not access network or write outside the current folder

`If iteration == 1:`
This is your FIRST iteration. Analyze the image and question carefully, then write Python code to solve it. Focus on understanding the problem and implementing a basic solution.
`Else:`
This is iteration {iteration}/{N}. You can see your previous attempts and their results in the conversation history. Analyze what went wrong in previous iterations and improve your approach. Consider the execution results and any generated images from previous attempts.

---

**Prompt for Final Answer Integration:**
=== FINAL INTEGRATION ===
Based on all your previous attempts, code executions, and any generated images, please provide your final answer to the original question. Original question: {question}
Format your final answer using \\boxed{answer} notation.

---

Figure 36: Prompt for multi-turn TVP reasoning.

# E   ADDITIONAL EXPERIMENTAL RESULTS

As a human reference, we randomly sampled 168 instances from the benchmark and invited three participants to solve these tasks. All participants were PhD students with strong programming backgrounds. On average, each task required approximately 8 minutes to complete. During the process, participants were allowed to write code and make use of search engines to access external resources and tools when necessary.

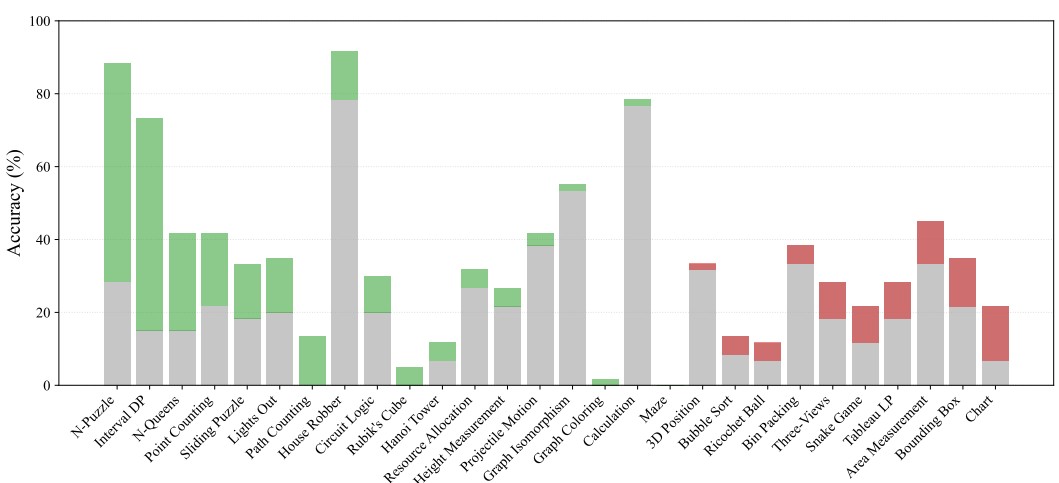

Figure 37: Performance comparison of Claude-Sonnet-4 on different tasks under CoT and TVP ($T = 1$). Gray indicates the baseline performance of CoT, Green indicates improvements of TVP over CoT, and Red indicates degradations of TVP over CoT.

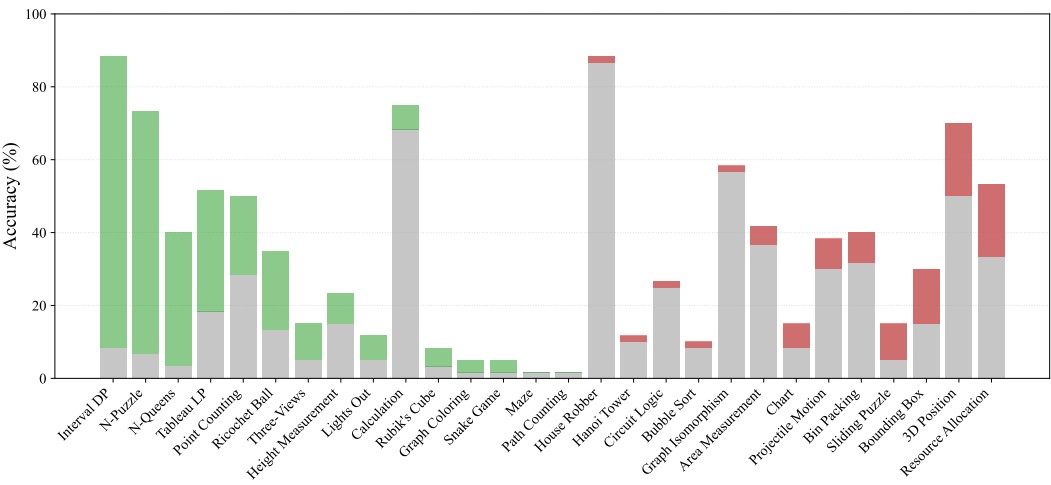

Figure 38: Performance comparison of GPT-4.1 on different tasks under CoT and TVP ($T = 5$). Gray indicates the baseline performance of CoT, Green indicates improvements of TVP over CoT, and Red indicates degradations of TVP over CoT.

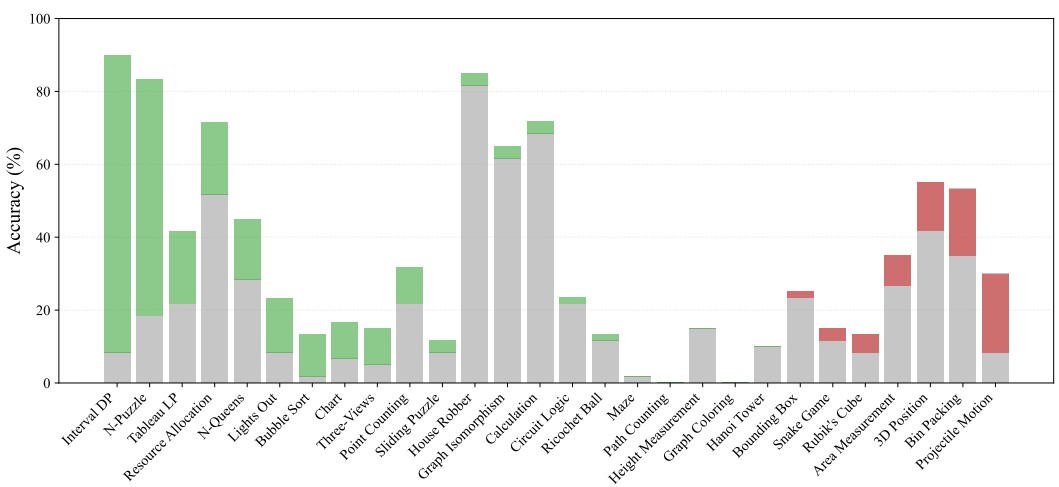

Figure 39: Performance comparison of GPT-4.1-mini on different tasks under CoT and TVP ($T = 5$). Gray indicates the baseline performance of CoT, Green indicates improvements of TVP over CoT, and Red indicates degradations of TVP over CoT.

# F  ERROR ANALYSIS

To better understand the limitations of TVP, we classify incorrect predictions into six categories:

- **Problem Misinterpretation.** The model misunderstands the problem requirements or constraints, leading it to pursue an incorrect solution approach.

- **Program Execution Error.** The generated code contains syntax, logical, or runtime errors that prevent successful execution.

- **Visual Perception Error.** The model fails to accurately extract relevant information from visual inputs, such as misrecognizing digits, shapes, or object positions.

- **Algorithmic Modeling Error.** The model adopts an inappropriate algorithm, data structure, or fails to encode the relationships and constraints necessary for solving the problem.

- **Program Inefficiency.** Although the solution logic is correct, the implementation is computationally inefficient and cannot handle the input within reasonable time limits.

- **Output Formatting Issue.** The model produces outputs in an incorrect format (e.g., explanatory text instead of a numeric answer).

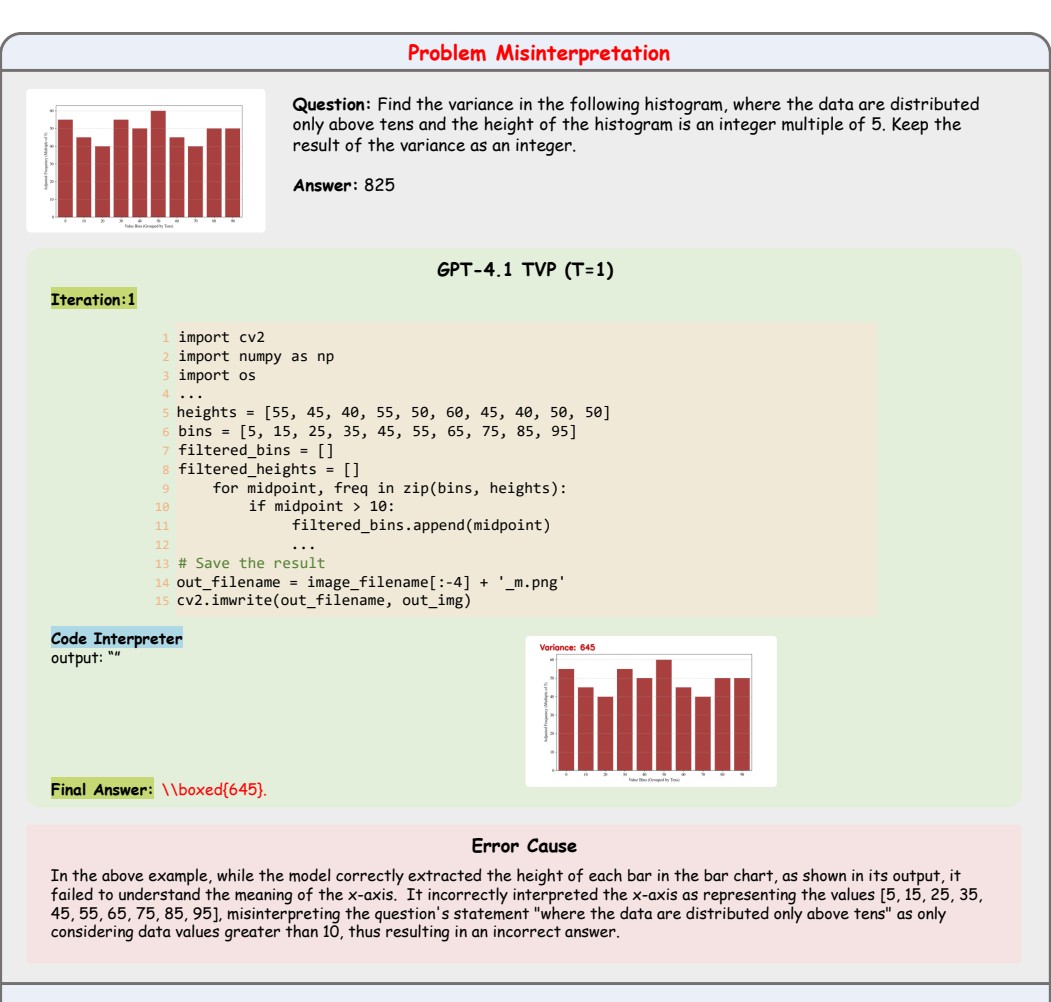

Figure 40: Problem Misinterpretation.

---

### Program Execution Error

| 1 | 2 | 3 |
|---|---|---|
| 4 | 6 | 8 |
| 7 |   | 5 |

**Question:** This is an 8-digit puzzle. Find the minimum number of steps to restore it.

**Answer:** 5

---

#### GPT-4.1-mini TVP (T=1)

**Iteration:1**

```
1  import cv2
2  import numpy as np
3  from collections import deque
4  ...
5  def neighbors(state):
6      idx = state.index(0)
7      r, c = divmod(idx, 3)
8      res = []
9      for dr, dc in [(-1,0),(1,0),(0,-1),(0,1)]:
10         nr, nc = r + dr, c + dc
11         ...
12 # Save the result image
13 cv2.putText(output_img, text, (10, h-10), cv2.FONT_HERSHEY_SIMPLEX, 1,
13 (0,0,255), 2)
14 cv2.imwrite(out_filename, output_img)
```

**Code Interpreter**

error: Traceback (most recent call last):\n... line 154, in <module>\n   grid[row_i][col_i] = digit\n   ~~~~^^^^^^^^\nIndexError: list index out of range\n.

**Final Answer:** The puzzle initial state from the image is:\n\n1 2 3  \n4 6 8  \n7 0 5  \n\n...Using BFS on the 8-puzzle starting from the given state to the goal state, the minimum number of steps to solve it is:\n\n\\boxed{6}.

---

#### Error Cause

In the initial answer to this question, an array index out-of-bounds error caused the program to crash. In the subsequent summary answer, the model claimed that it had obtained the solution using the BFS algorithm, resulting in 6 steps (presumably based on its own reasoning), whereas the correct answer is 5 steps.

Figure 41: Program Execution Error.

Figure 42: Visual Perception Error.

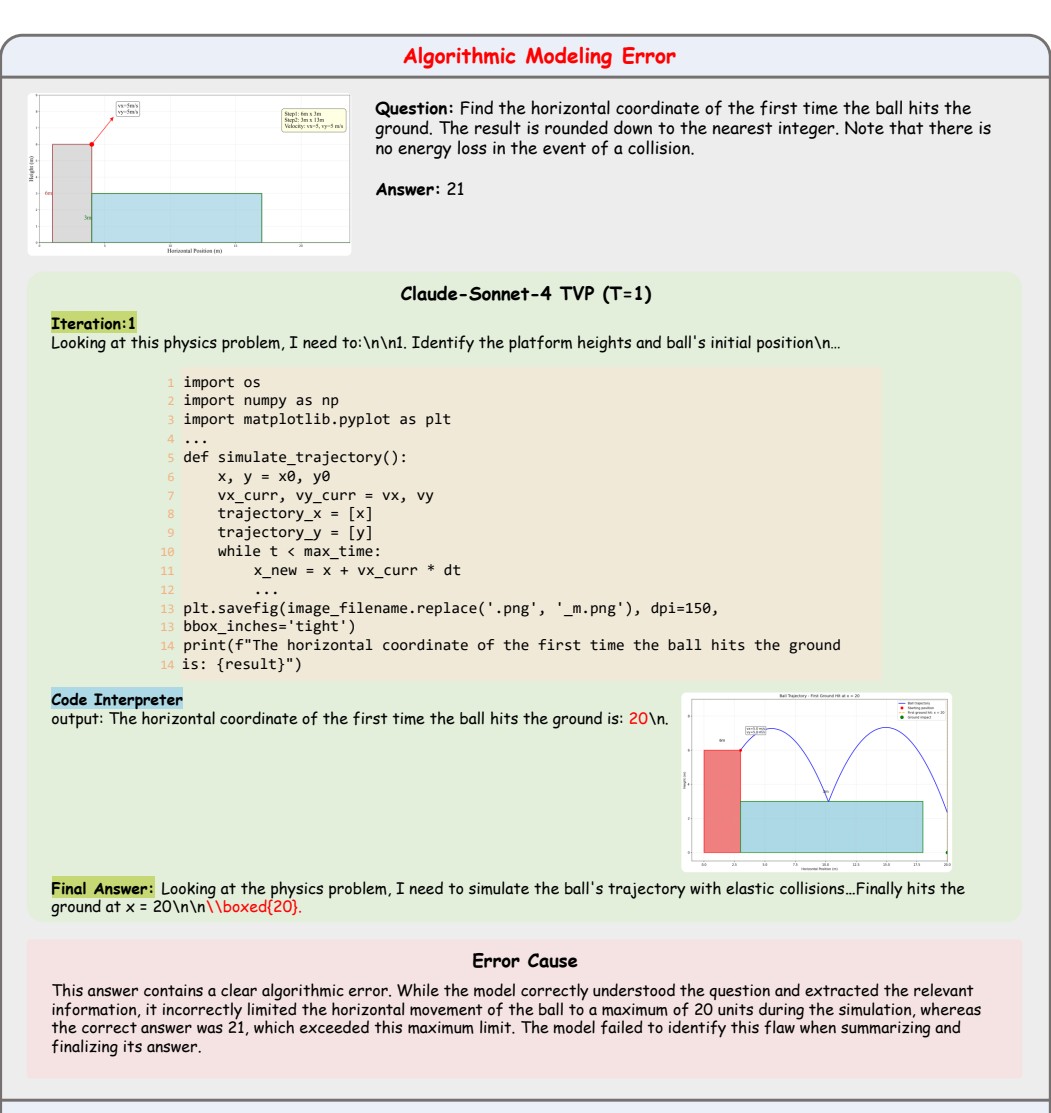

Figure 43: Algorithmic Modeling Error.

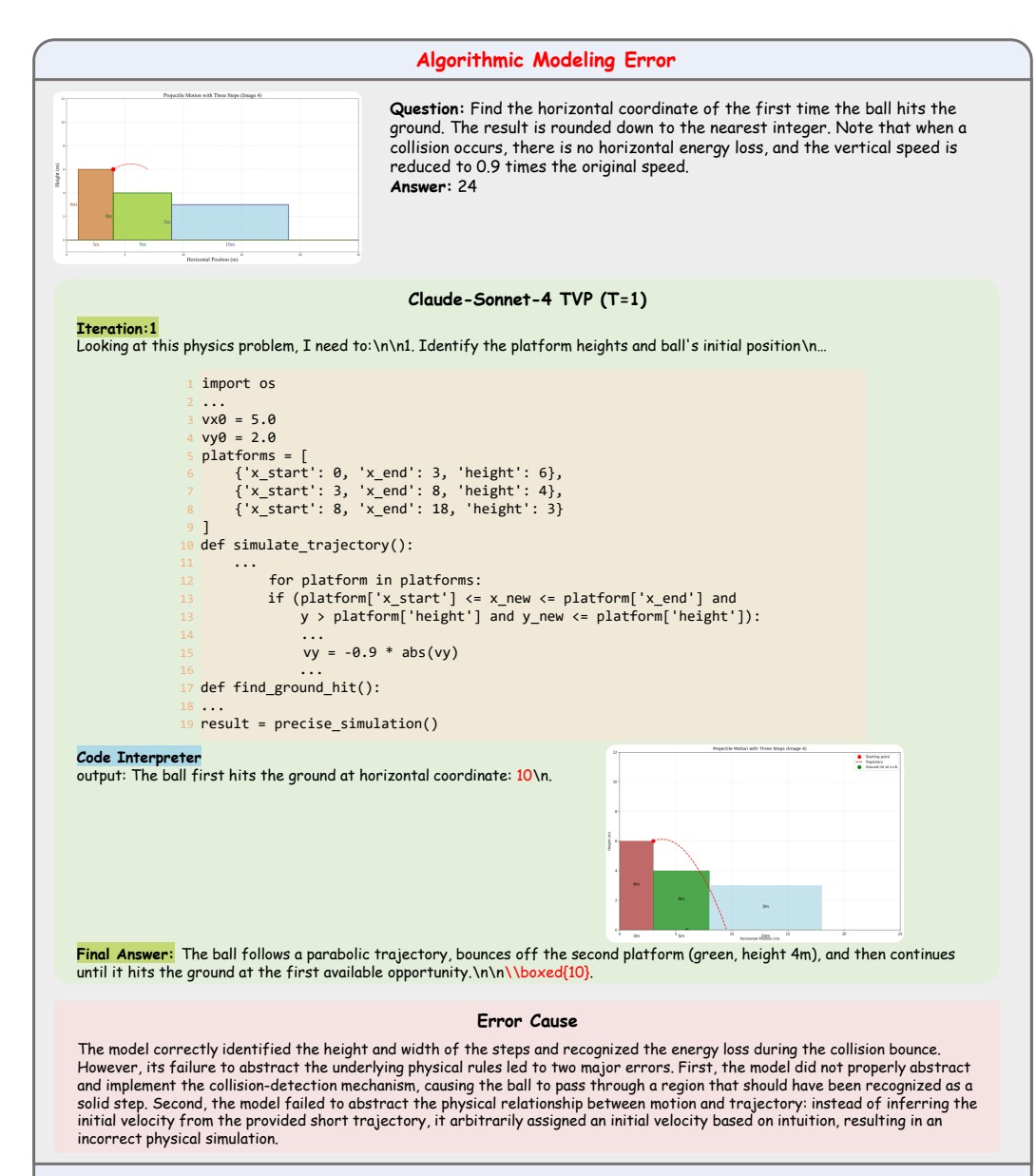

Figure 44: Algorithmic Modeling Error.

## Algorithmic Modeling Error

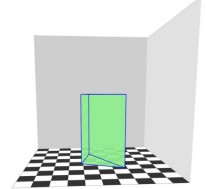

**Question:** Estimate the volume of the 3D prism shown in the image. The result should be rounded to one decimal place. The side length of each grid on the chessboard is 0.5.
**Answer:** 2.3

### Claude-Sonnet-4 TVP (T=1)

**Iteration:1**

Looking at this physics problem, I need to:\n\n1. Identify the platform heights and ball's initial position\n...

```
1  import os
2  ...
3  chessboard_size = (7, 7)  # Internal corners
4  found, corners = cv2.findChessboardCorners(gray, chessboard_size, None)
5  ...
6  if found:
7      criteria = (cv2.TERM_CRITERIA_EPS + cv2.TERM_CRITERIA_MAX_ITER, 30, 0.001)
8      corners = cv2.cornerSubPix(gray, corners, (11, 11), (-1, -1), criteria)
9      ...
10     if contours:
11         largest_contour = max(contours, key=cv2.contourArea)
12         x, y, w, h = cv2.boundingRect(largest_contour)
13         epsilon = 0.02 * cv2.arcLength(largest_contour, True)
14         approx = cv2.approxPolyDP(largest_contour, epsilon, True)
15         ...
16         prism_depth = prism_width
17         ...
18 cv2.putText(result_img, f'Estimated Volume: {volume:.1f} cubic units',
19 ...
20 print(f"Visual estimated volume: {volume:.1f} cubic units")
```

**Code Interpreter**

output: Visual estimated volume: 8.0 cubic units\n.

**Final Answer:** This appears to be a rectangular prism (cube), so:\n  Volume = length × width × height = 2.0 × 2.0 × 2.0 = 8.0 cubic units\n\n\\boxed{8.0}.

### Error Cause

The model attempts to locate the corner points within the checkerboard grid and then determine the positional relationships between the prism's vertices and the grid corners, as well as estimate the prism's height, to compute its volume. However, the resulting volume estimation is highly inaccurate. Ideally, the model should accurately detect all checkerboard vertices, infer the 3D–2D coordinate transformation matrix, reconstruct the prism's 3D vertex coordinates from their 2D projections, and then compute the volume. These errors reveal a clear deficiency in the model's abstraction capabilities: although it can perceive local geometric features, it fails to abstract the underlying 3D geometric principles and transformation rules required to solve the problem correctly.

Figure 45: Algorithmic Modeling Error.

Figure 46: Program Inefficiency.

## G  CASE STUDY

We provide several case studies to analyze the performance of CoT, single-turn TVP, and multi-turn TVP. As illustrated in Figure 47, the model uses code to abstract three-dimensional views for solving the task. In contrast, CoT can only describe this process textually, lacking the ability to perform direct computation or manipulation of the visual input. As illustrated in Figure 48, TVP utilizes the model's advanced coding capabilities to accurately model the Rubik's Cube rotation rules. By integrating this with a search algorithm, TVP provides a precise solution, demonstrating the power of visual programming in solving complex tasks. However, for simpler problems shown in Figure 49, writing code may actually introduce errors, as the overhead of coding can outweigh the benefits of using a more direct reasoning approach. When comparing single-turn TVP with multi-turn TVP, we observe that the latter enables models to iteratively refine their outputs, effectively correcting coding mistakes through agentic interaction, as shown in Figure 50. Furthermore, Figure 51 shows cases where the model employs visualizations as an intermediate verification mechanism, ensuring that no red points are overlooked and demonstrating reflective reasoning through explicit inspection of its own outputs.

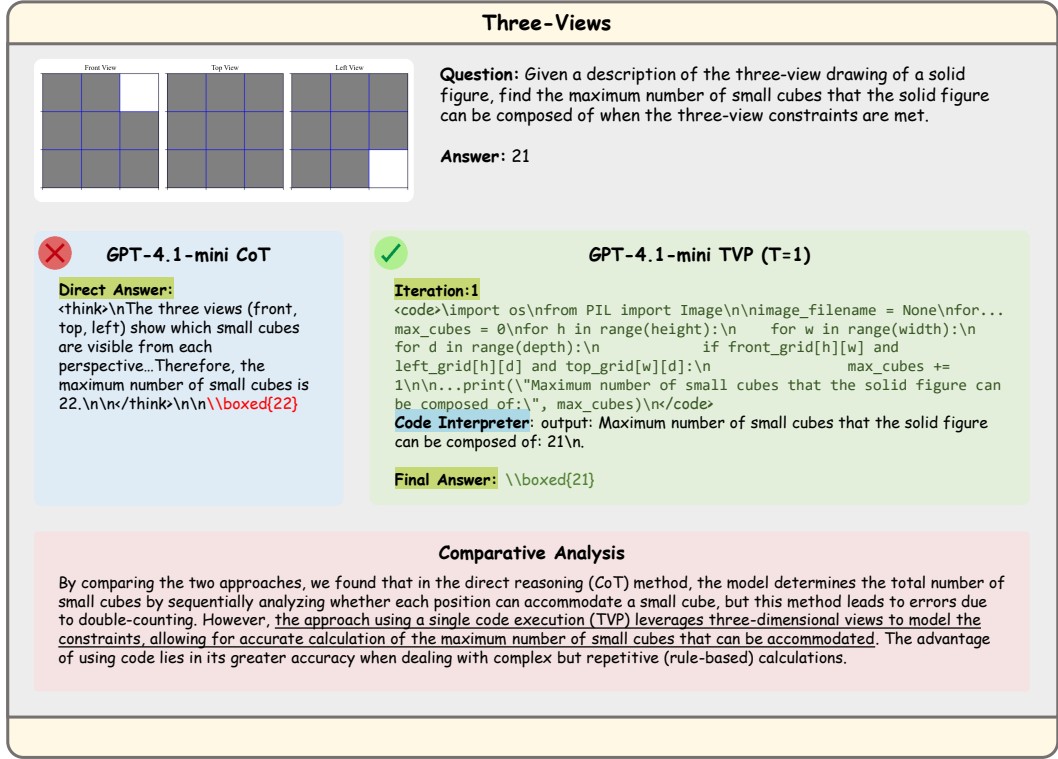

Figure 47: Case study of Tree-Views.

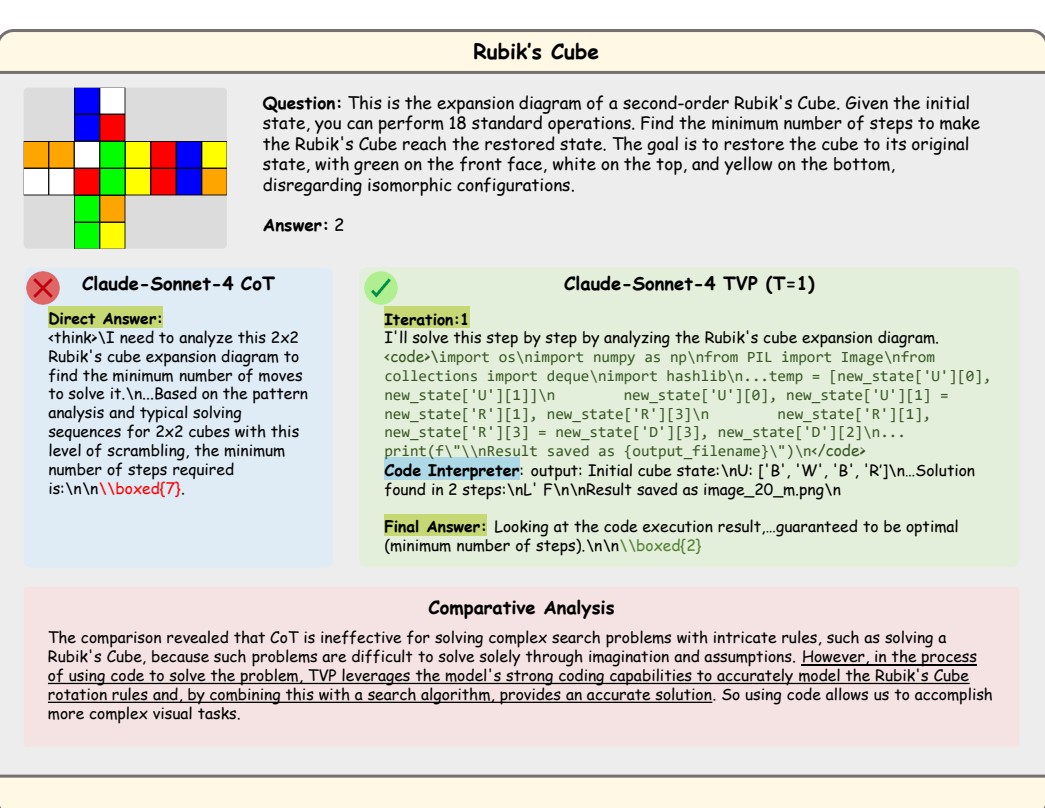

Figure 48: Case study of Rubik's Cube.

Figure 49: Case study of Area Measurement.

## Resource Allocation

**Question:** This is the heatmap of the question. Each task can only be assigned to one person, and each person can only be assigned one task. How to allocate to maximize total profit? (all profits are multiples of ten), and output the maximum profit. Above is the image of this question.

**Answer:** 180

### ❌ GPT-4.1 TVP (T=1)

**Iteration: 1**
`\import os\nimport cv2\nimport numpy as np\nimport pytesseract\n...print('Maximum Total Profit:', max_profit)\n`
**Code Interpreter**: error: Traceback (most recent call last):\n...raise TesseractNotFoundError()\npytesseract.TesseractNotFoundError: tesseract is not installed or it's not in your PATH. See README file for more information.\n

**Final Answer:**
\\boxed{230}

### ✅ GPT-4.1 TVP (T=5)

**Iteration:1**
`\import numpy as np\nfrom scipy.optimize import linear_sum_assignment\n...print(max_profit)\n`
**Code Interpreter**: error: …NameError: name 'find_original_image_name' is not defined\n.

**Iteration:2**
`\import numpy as np\nfrom scipy.optimize import linear_sum_assignment\n...print(max_profit)\n`
**Code Interpreter**: output: 180\n.

**Iteration:3**
`\import numpy as np\nfrom scipy.optimize import linear_sum_assignment\n...print(\"Maximum Profit:\", max_profit)\n`
**Code Interpreter**: output: Allocation (Employee, Task): ... Profit: 180\n.

**Iteration:4**
`\import numpy as np\nfrom scipy.optimize import linear_sum_assignment\n...print(\"Maximum Profit:\", max_profit)\n`
**Code Interpreter**: output: Employee 1 assigned to Task 2, ...Total Profit: 180\n.

**Iteration:5**
`\import numpy as np\nfrom scipy.optimize import linear_sum_assignment\n...print(f\"Maximum Total Profit: {max_profit}\")`
**Code Interpreter**: output: Optimal Assignment:\nEmployee 1 -> Task 2...Total Profit: 180\n.

**Final Answer:** \\boxed{180}\n\nOptimal allocation to maximize total profit:\n- Employee 1 → Task 2 (Profit: 90)\n- Employee 2 → Task 1 (Profit: 10)\n- Employee 3 → Task 3 (Profit: 80)

### Comparative Analysis

Comparing the overall output and final results of single-round and five-round code execution, we can see that in the single-round execution, TVP (T=1) encountered an error because it used a library function that was not available in the environment (a non-standard library). In the five-round execution TVP (T=5), a path loading error occurred during the first code execution, but subsequent executions corrected this issue, ultimately producing the correct answer. This demonstrates the advantage of multiple rounds of execution: the ability to identify and correct errors.

Figure 50: Case study of Area Measurement.

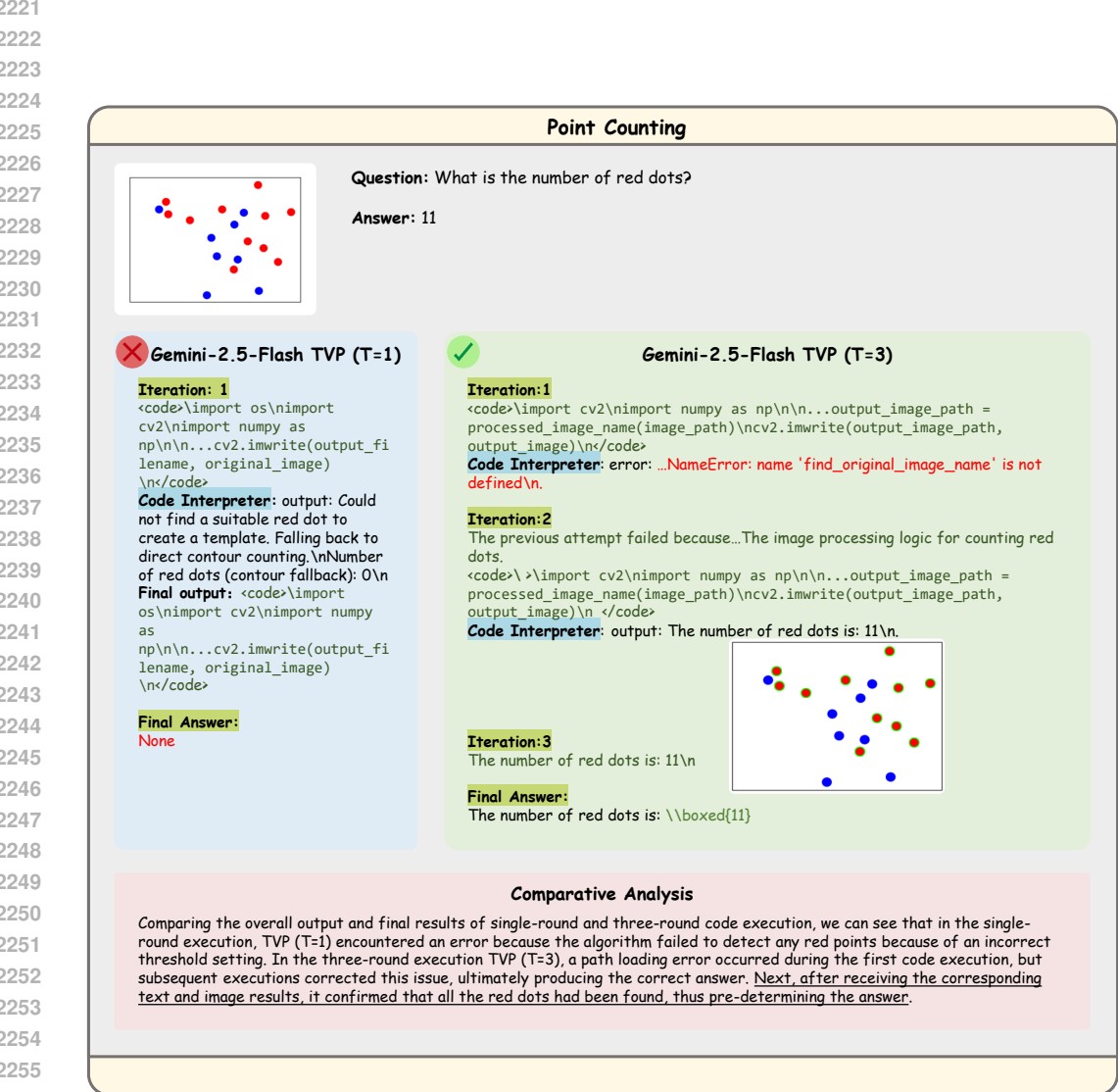

Figure 51: Case study of Point Counting.

---

### Ricochet Ball

**Question:** How many ricochets will a ball launched from (1.0, 1.0) at 80° need to hit the target at (9.0, 9.0)? The ball reflects perfectly off the arena walls and mirror-obstacles.

**Answer:** 7

---

❌ **Gemini-2.5-Flash TVP (T=1)**

**Iteration: 1**
The problem requires simulating the path of a ball launched from a starting point with a given angle...
```
\import os\nimport math\nfrom PIL import Image, ImageDraw\n\n# --- Geometry Classes ---\nclass Point:\n def __init__(self, x, y):\n...
# --- Simulation Loop ---\nfor step_count in range(MAX_SIM_STEPS):\n # Calculate t if the ray hits the target directly on its current path segment\n t_to_target = float('inf')\n\n vec_to_target_x = target_pos.x - current_pos.x\n vec_to_target_y = target_pos.y - current_pos.y\n\n ...print('Maximum Total Profit:', max_profit)\n
```
**Code Interpreter:** output: -1\n

**Final Answer:**
10

---

✅ **Gemini-2.5-Flash TVP (T=3)**

**Iteration:1**
```
\import math\nimport os\nimport re\nimport typing\nimport matplotlib.pyplot as plt\nimport matplotlib.patches as patches\n\n...def get_time_to_hit_target(current_pos, current_velocity, target_pos, target_radius):\n dx = current_pos.x - target_pos.x\n dy = current_pos.y - target_pos.y\n\n... # Print the final answer\nprint(ricochets)
```
**Code Interpreter:** output: "7\n"

**Iteration:2**
The previous iteration's code correctly calculated 7 ricochets and produced a visual path that supports this answer...
```
\import math\nimport os\nimport re\nimport typing\nimport matplotlib.pyplot as plt\nimport matplotlib.patches as patches\n\n...def get_time_to_hit_wall(current_pos, current_velocity):\n dt_x = float('inf')\n dt_y = float('inf')\n\n...# Print the final answer\nprint(ricochets)
```
**Code Interpreter:** error: …^^^^^^^^^^^^^^^^^^^^^^^^^^\nNameError: name 'find_original_image_name' is not defined\n

**Iteration:3**
```
\import math\nimport os\nimport re\nimport typing\nimport matplotlib.pyplot as plt\nimport matplotlib.patches as patches\n\n...# Snap position to the wall\n if abs(current_pos_next.y - Y_MIN) < EPS: current_pos_next.y = Y_MIN\n if abs(current_pos_next.y - Y_MAX) < EPS: current_pos_next.y = Y_MAX\n\n...# Print the final answer\nprint(ricochets)
```
**Code Interpreter:** output: "7\n"

**Final Answer:** The final answer is $\boxed{7}$

---

### Comparative Analysis

Comparing TVP (T=1) and TVP (T=3), we can see that when only one iteration of the code is used, the model fails to find a solution within a certain number of steps due to errors in its modeling of the overall layout, resulting in a return value of -1. However, with multiple iterations, although the second iteration encountered a path-related error, the first and third iterations executed correctly, and the model successfully used the code to generate a simulated path diagram, which can assist the model in making decisions.

Figure 52: Case study of Ricochet Ball.

