# OpenReview forum: "Pixels Lie, Code Doesn't: Thinking with Visual Programming for ''Seemingly Impossible'' Multimodal Agentic Reasoning Tasks"
_ICLR.cc/2026/Conference — Submitted to ICLR 2026_

### Official Review · Reviewer_TRk7 · 2025-10-31

**Soundness:** 2
**Presentation:** 2
**Contribution:** 2
**Rating:** 4
**Confidence:** 3

**Summary:**

The paper investigates the limitations of Chain-of-Thought (CoT) and proposes methods to improve the upper bound of multimodal reasoning. They involve a external code executor to iteratively generate, run, and verify both visual and textual agentic operations. Comprehensive findings show the deficiency and performance pattern in current TVP models.

**Strengths:**

The researched question of Multi-modal Reasoning plays a significant role in multimodal models, which endorses the model's thinking capabilities.

External Code Executor is an efficient tool that has demonstrated efficiency in multiple settings.

The author conducted a comprehensive experiment that shows the deficiency of current TVP models.

**Weaknesses:**

Some typos need to be revised, such as OpenAI’s o4 may likely be OpenAI’s 4o.

The difficult ladder is more likely designed manually, and lacks the theory to reflect the fundamental difference. This deserves thinking about the method for the difficulty ladder, which may start from basic cognitive capability.

How to ensure the quality and label correctness of the benchmark?

As shown in Table 2, the best MLLMs have a great gap from humans. What is the root cause behind this phenomenon? And what training or training-free method has the potential to improve the performance on these tasks?

It will be better if you provide some failure cases and visual examples across different baselines and proposed methods.
Is the data curation possible to transfer to real CV or robot tasks?

**Questions:**

See Weakness.

---

> ### Author Response · Authors · 2025-11-15
> **Rebuttal 1**
>
> Thank you for your thoughtful and detailed feedback. We sincerely hope that our response has effectively addressed your concerns.
>
> > Some typos need to be revised, such as OpenAI’s o4 may likely be OpenAI’s 4o.
>
> We would like to clarify that **we indeed intended to refer to [OpenAI's o4 model](https://openai.com/index/introducing-o3-and-o4-mini/)**, which is a version that natively supports agentic tool use. Unlike 4o, o4 can agentically utilize and combine internally available tools, significantly enhancing its ability to perform complex tasks.
>
>
> > The difficult ladder is more likely designed manually, and lacks the theory to reflect the fundamental difference. This deserves thinking about the method for the difficulty ladder, which may start from basic cognitive capability.
>
> Thank you for your insightful comment. In the MMR-VIP framework, **we have specifically designed the difficulty ladder from the perspectives of cognitive capabilities and computational complexity theory**. The problems are categorized into three levels:
>
> 1. **Easy level**: Tasks that the model can solve independently using its inherent perception and reasoning abilities, without programming assistance. These problems correspond to "low-complexity problems in P" where the model can reason within its working memory. From a cognitive capabilities perspective, **the Easy level tasks correspond to problems that require minimal cognitive load to solve**. For example, in tasks like Path Counting, the number of possible paths in easy-difficulty problems typically follows $O(10)$, which can be easily counted by the model.
>
> 2. **Medium level**: Problems that are challenging for the model to solve on its own but can be effectively tackled with the help of a code interpreter. These problems are typically "polynomial-time solvable problems in P" requiring external tools to compute solutions. From a cognitive capabilities perspective, **these problems occupy a significant amount of working memory and typically involve computational demands that exceed the capabilities of existing models**. These capabilities are not inherently gained simply by increasing the model's size or the volume of training data. For example, in tasks like Path Counting, the number of possible paths in medium-difficulty problems typically follows $O(10^3)$. If the model were to rely solely on brute-force deep search within the text space, it would struggle to compute the paths accurately due to the inherent complexity and the length of the reasoning process. This challenge is further compounded when the reasoning involves long token sequences, as the model's working memory and computational capacity would be insufficient to handle such extensive computations without external tools.
>
> 3. **Hard level**: Problems that remain unsolved even with programming assistance, often due to large-scale computational complexity, intricate constraints, or demanding optimization. These problems are conceptually "NP-hard" and they are inherently difficult both for humans and for models. From a cognitive capabilities perspective, **these tasks require more advanced cognitive processes that go beyond the model’s basic reasoning abilities, necessitating the development of innovative problem-solving strategies to obtain feasible solutions.** *For example, in tasks like Path Counting, the number of possible paths in hard-difficulty problems typically follows $O(10^7)$. Using a conventional depth-first search algorithm alone would lead to timeouts due to the computational complexity. Therefore, an effective strategy is to divide the path into segments, perform depth-first search on each segment separately, and then combine the partial results, which significantly improves overall efficiency.*
>
> Thus, our difficulty ladder is indeed designed based on cognitive capabilities and computational complexity theory, reflecting the inherent challenges in both reasoning and computational resources required to solve tasks at different difficulty levels.

---

> ### Author Response · Authors · 2025-11-15
> **Rebuttal 2**
>
> > How to ensure the quality and label correctness of the benchmark?
>
> **To ensure the quality and label correctness of the benchmark, the entire dataset is generated using a Code2Task generation framework.** In this framework, annotators are responsible for writing code that specifies task rules and automatically generates the corresponding images, problems, and answers. As the benchmark is fully code-generated, the focus is primarily on ensuring the correctness of the underlying code logic. **As long as the code logic is correct, the generated tasks, images, and answers are inherently reliable and free from labeling errors.**
>
> To further guarantee the accuracy of the generated content, **we performed cross-validation of the code, where each program was independently reviewed by multiple annotators to confirm its correctness**. Additionally, we designed several test cases to rigorously validate the functionality and correctness of the code.
>
> > As shown in Table 2, the best MLLMs have a great gap from humans. What is the root cause behind this phenomenon? And what training or training-free method has the potential to improve the performance on these tasks?
>
> Thank you for your insightful question. **The observed performance gap between the best MLLMs and humans can largely be attributed to the superior tool-use capabilities of humans.** In our study, each participant was allowed to leverage external tools such as search engines and code interpreters during the process. For example, in tasks like solving a Rubik’s Cube puzzle, participants could use tools such as [Rubik’s Cube Solver](https://rubiks-cube-solver.com/) to aid in solving the problem efficiently. This access to external resources significantly enhances human performance, enabling them to overcome the cognitive limitations that MLLMs currently face.
>
> As a training-free method to improve MLLM performance, **one potential approach is to allow models to use search engines in addition to programming tools**. However, this strategy would primarily benefit models with strong tool-use capabilities, such as GPT-5, which can more effectively integrate external resources during the task-solving process.
>
> Regarding training methods, we need to first enhance the model's visual programming capabilities through SFT. Bridging this gap requires **equipping models with stronger visual programming capabilities and more advanced visual abstraction skills**. On this foundation, agentic reinforcement learning can further enhance the model's ability to perform multi-turn code invocation.
>
> > It will be better if you provide some failure cases and visual examples across different baselines and proposed methods.
>
> Thank you for your suggestion. **We have included error analysis in Appendix F and case study in Appendix G.** In these sections, we not only analyze the causes of errors but also provide a comparison of the outputs from the original CoT and TVP (for $T = 1, 3, 5$).
>
> 1. As illustrated in Figure 45, the model uses code to abstract three-dimensional views for solving the task. In contrast, CoT can only describe this process textually, lacking the ability to perform direct computation or manipulation of the visual input.
>
> 2. As shown in Figure 46, TVP utilizes the model’s advanced coding capabilities to accurately model the Rubik’s Cube rotation rules. By integrating this with a search algorithm, TVP provides a precise solution, demonstrating the power of visual programming in solving complex tasks.
>
> 3. However, for simpler problems shown in Figure 47, writing code may actually introduce errors, as the overhead of coding can outweigh the benefits of using a more direct reasoning approach.
>
> 4. When comparing single-turn TVP with multi-turn TVP, we observe that the latter enables models to iteratively refine their outputs, effectively correcting coding mistakes through agentic interaction, as shown in Figure 48.
>
> 5. Furthermore, Figure 49 shows cases where the model employs visualizations as an intermediate verification mechanism, ensuring that no red points are overlooked and demonstrating reflective reasoning through explicit inspection of its own outputs.
>
> > Is the data curation possible to transfer to real CV or robot tasks?
>
> Thank you for your question. The Code2Task approach we use can indeed be transferred to real robot tasks. For example, by utilizing code generation, **we can synthesize robotic scenarios such as object manipulation, path planning, or autonomous navigation**. This approach allows us to generate a wide variety of task scenarios that closely mimic real-world robotic challenges, making it suitable for training robotic models.

---

> ### Author Response · Authors · 2025-11-23
> **Looking forward to your feedback!**
>
> Dear Reviewer TRk7,
>
> Thank you once again for your valuable feedback. We have added further explanations and clarifications to the paper based on your comments. Since the discussion phase has already begun, we would greatly appreciate knowing whether our responses have addressed your concerns. Your insights are extremely valuable to us, and we are very willing to address any remaining issues to further improve the work.

---

> ### Author Response · Authors · 2025-11-27
> **Looking forward to further discussion with you.**
>
> Dear Reviewer TRk7,
>
> Thank you once again for your valuable feedback. We have added further explanations and clarifications to the paper based on your comments. We sincerely hope to clarify the misunderstandings and address any concerns you may have. We have clarified that o4 is not a typo, introduced the principles behind the design of the difficulty ladder, and explained how the correctness of the benchmark is ensured. Additionally, we would like to emphasize that we have provided error analysis in Appendix F and case studies in Appendix G in the original version. Your insights are extremely valuable to us, and we would be very grateful to hear whether there is any possibility of improving the score at this stage.

---

### Official Review · Reviewer_9r1P · 2025-10-31

**Soundness:** 2
**Presentation:** 3
**Contribution:** 3
**Rating:** 4
**Confidence:** 3

**Summary:**

This paper introduces MMR-VIP, a benchmark for multimodal reasoning under the proposed paradigm of Thinking with Visual Programming (TVP). It evaluates how models use code execution to solve visual and symbolic reasoning problems across 28 task types, three difficulty levels, and three cognitive skills. The authors assess a range of models, showing that TVP improves performance over CoT reasoning, especially on medium-difficulty tasks. Overall, the paper aims to establish a new framework for studying executable multimodal reasoning.

**Strengths:**

- **S1.** The benchmark provides a large dataset that tests visual reasoning with visual programming, providing a useful resource for future research.
- **S2.** The benchmark covers 28 task types across three difficulty levels and three cognitive skills, could be very useful for the community to evaluate MLLMs. Several tasks are also quite interesting (e.g., physics, network).
- **S3.** The paper is written well and very easy to follow.

**Weaknesses:**

- **W1.** The proposed difficulty ladder is somehow problematic. Its definition depends on current model performance rather than on intrinsic task properties, making it unstable and model-dependent. For example, the 'hard' level is defined by whether existing models can solve it with programming assistance; as models improve, tasks once labeled 'hard' might shift to 'medium.'
- **W2.** The three cognitive skill dimensions (Perception, Abstraction, Optimization) are intuitively reasonable but not theoretically or empirically justified. If these dimensions are claimed as a main contribution, they should be either theoretically grounded or empirically validated to demonstrate their distinctiveness and usefulness. It is fine to include such categorization for analytical purposes, but presenting it as a core contribution requires stronger justification.
- **W3.** Section 3.3 lacks transparency in benchmark construction: the paper didn't provide much information on how annotators were recruited, what initial materials or code templates they were given, and whether they followed a unified guideline or independently designed tasks. It also remains unclear how the authors ensured diversity across annotators, how quality control and validation were conducted. I hope the author could provide more details about the benchmark construction.
- **W4.** Although the paper claims to include 1680 tasks, these are derived from only 28 seed task types x 3 difficulty levels x 20 instances per combination, which indicates limited underlying diversity. This structure suggests that most tasks may share similar templates or reasoning structures, differing mainly by parameter changes.
- **W5.** The related work section omits discussion of prior works on visual programming. For example, [1] is also a visual programming benchmark and uses similar code2Task method for data generation. [2] also considers the visual programming setting, in which they synthesize code for visual reasoning. These visual programming work are quite related but not discussed.

References:

[1] Program Synthesis Benchmark for Visual Programming in XLogoOnline Environment. ACL 2025.

[2] Visual Programming: Compositional visual reasoning without training. CVPR 2023

**Questions:**

- **Q1.** In Table 2, why are some open-source models evaluated with T=1 while others include T=3?
- **Q2.** What exactly defines a native TVP model? Why are such models not evaluated under CoT, T=1, or T=3?
- **Q3.** How do the authors ensure that the dataset instances represent genuinely distinct reasoning challenges rather than minor parameter variations?
- **Q4**. How did you come up with these 28 task types? Is it motivated by any existing benchmarks, designed from scratch, or other cases?

**Details Of Ethics Concerns:**

The paper uses human participants for dataset construction and evaluation but doesn’t explain their involvement procedure or compensation. It’s unclear whether they were paid, how much, or if consent was collected.

---

> ### Author Response · Authors · 2025-11-18
> **Rebuttal 1**
>
> Thank you for your thoughtful and detailed feedback. We sincerely hope that our response has effectively addressed your concerns.
>
> > W1: The proposed difficulty ladder is somehow problematic. Its definition depends on current model performance rather than on intrinsic task properties, making it unstable and model-dependent. For example, the 'hard' level is defined by whether existing models can solve it with programming assistance; as models improve, tasks once labeled 'hard' might shift to 'medium'.
>
> Thank you for the thoughtful comment. We would like to clarify that our difficulty ladder is not defined by current model performance, but by intrinsic task properties grounded in cognitive capabilities and computational complexity theory. The problems are categorized into three levels:
>
> 1. **Easy level**: Tasks that the model can solve independently using its inherent perception and reasoning abilities, without programming assistance. These problems correspond to "low-complexity problems in P" where the model can reason within its working memory. From a cognitive capabilities perspective, **the Easy level tasks correspond to problems that require minimal cognitive load to solve**. *For example, in tasks like Path Counting, the number of possible paths in easy-difficulty problems typically follows $O(10)$, which can be easily counted by the model.*
>
> 2. **Medium level**: Problems that are challenging for the model to solve on its own but can be effectively tackled with the help of a code interpreter. These problems are typically "polynomial-time solvable problems in P" requiring external tools to compute solutions. From a cognitive capabilities perspective, **these problems occupy a significant amount of working memory and typically involve computational demands that exceed the capabilities of existing models**. These capabilities are not inherently gained simply by increasing the model's size or the volume of training data. *For example, in tasks like Path Counting, the number of possible paths in medium-difficulty problems typically follows $O(10^3)$. If the model were to rely solely on brute-force deep search within the text space, it would struggle to compute the paths accurately due to the inherent complexity and the length of the reasoning process.* This challenge is further compounded when the reasoning involves long token sequences, as the model's working memory and computational capacity would be insufficient to handle such extensive computations without external tools.
>
> 3. **Hard level**: Problems that remain unsolved even with programming assistance, often due to large-scale computational complexity, intricate constraints, or demanding optimization. These problems are conceptually "NP-hard" and they are inherently difficult both for humans and for models. From a cognitive capabilities perspective, **these tasks require more advanced cognitive processes that go beyond the model’s basic reasoning abilities, necessitating the development of innovative problem-solving strategies to obtain feasible solutions.** *For example, in tasks like Path Counting, the number of possible paths in hard-difficulty problems typically follows $O(10^7)$. Using a conventional depth-first search algorithm alone would lead to timeouts due to the computational complexity. Therefore, an effective strategy is to divide the path into segments, perform depth-first search on each segment separately, and then combine the partial results, which significantly improves overall efficiency.*
>
> Thus, our difficulty ladder is indeed designed based on cognitive capabilities and computational complexity theory, relying on the intrinsic structural properties of each task. **Consistent with this design, our human study results in Table 2 demonstrate a clear difficulty gap, with the hard level posing substantial challenges even to human annotators.** Naturally, as models continue to improve, tasks that are currently designed as hard may shift toward the medium category, and medium tasks may eventually become easy. This phenomenon is common across benchmarks. For example, the AIME benchmark was once considered difficult but has become relatively easy as model capabilities have advanced. **Nevertheless, the relative ordering of our difficulty ladder remains stable because it reflects intrinsic task structure rather than current model performance.**

---

> > ### Author Response · Authors · 2025-11-18
> > **Rebuttal 2**
> >
> > > W2: The three cognitive skill dimensions (Perception, Abstraction, Optimization) are intuitively reasonable but not theoretically or empirically justified. If these dimensions are claimed as a main contribution, they should be either theoretically grounded or empirically validated to demonstrate their distinctiveness and usefulness. It is fine to include such categorization for analytical purposes, but presenting it as a core contribution requires stronger justification.
> >
> > We appreciate the reviewer’s concern regarding the theoretical grounding of our three cognitive skill dimensions. We would like to clarify that **the proposed dimensions, Perception, Abstraction, and Optimization, are indeed directly supported by well-established theories in cognitive science.**
> >
> > Specifically, our formulation follows the **Perception–Cognition–Action (PCA)** model [1, 2], a foundational framework in cognitive psychology that characterizes intelligent behavior as a sequential process of **(1) perceiving sensory inputs, (2) forming internal representations and abstractions, (3) selecting and executing goal-directed actions**. This mapping aligns naturally with our three dimensions:
> >
> > 1. Perception corresponds to the first stage of the PCA framework, where raw sensory information is interpreted and transformed into meaningful signals.
> >
> > 2. Abstraction corresponds to the cognitive processing stage, where internal representations and relational structures are constructed.
> >
> > 3. Optimization corresponds to the action stage, where agents evaluate alternatives and select strategies to achieve task objectives, consistent with theories of goal-directed decision-making and executive function.
> >
> > Overall, our framework is theoretically grounded in established cognitive models, and empirically, we also observe that models consistently exhibit these three skills within the TVP paradigm.
> >
> > [1] Upper processing stages of the perception–action cycle. Fuster, Joaquı́n M. Trends in Cognitive Sciences.
> >
> > [2] Neisser U. Cognitive psychology: Classic edition[M]. Psychology press, 2014.

---

> ### Author Response · Authors · 2025-11-18
> **Rebuttal 3**
>
> > W3: Section 3.3 lacks transparency in benchmark construction: the paper didn't provide much information on how annotators were recruited, what initial materials or code templates they were given, and whether they followed a unified guideline or independently designed tasks.
>
> We apologize for not providing sufficient details regarding the benchmark construction process. **We would like to clarify that all human annotations in this study were conducted by five annotators who are also co-authors of the paper. All annotators are AI or CS students with substantial programming experience and domain knowledge relevant to the tasks. Participation was voluntary, and compensation was provided in the form of standard research assistant payments.**
>
> **In fact, we have already provided a concise annotation guideline in Appendix C (Page 16).** For completeness, we also include a full guideline that specifies the task construction principles, required task formats, difficulty definitions, and an N-Queens code example illustrating the data generation workflow.
>
> ```
> ## 1. Guiding Principles
>
> Our goal is to create a comprehensive benchmark that evaluates a model's multimodal agentic reasoning capabilities.
>
> All tasks must:
> (1) Be code-synthesizable (problems, images, and solutions are generated by code).
> (2) Be aligned with cognitive skills (at least 1, at most 3 from the given taxonomy).
> (3) Be stratified into difficulty levels (Easy/Medium/Hard).
> (4) Be suitable for programmatic reasoning (problems solvable or aided by code execution).
>
> ### 1.1. Difficulty Ladder
>
> We define three levels of difficulty based on the external tools required for a solution:
>
> (1) Easy: solvable using intrinsic perceptual and reasoning abilities, without code execution.
> (2) Medium: requiring programmatic operations, where external computation is essential.
> (3) Hard: remaining challenging even with programming support, typically due to high algorithmic complexity or intricate constraints.
>
> ### 1.2. Cognitive Capabilities
>
> We evaluate the models from three cognitive perspectives (Perception, Abstraction, Optimization), which are broken down into nine specific task types. Your generated problem must target one or more of these.
>
> (1) Attribute: identify colors, shapes, sizes.
> (2) Location: detect positions, distances, spatial relations.
> (3) Symbolic: recognize digits, letters, or visual symbols.
> (4) Geometry: formulate geometric equations or relations.
> (5) Physics: model dynamics using physical laws.
> (6) Network: construct graph structures (nodes, edges, constraints).
> (7) Search: implement DFS, BFS, or other exploration methods.
> (8) Planning: apply dynamic/linear programming to solve constrained problems.
> (9) Computation: perform numerical calculations or algorithmic procedures.
>
> ## 2. Standard Generation Workflow
>
> All tasks are generated programmatically to ensure scalability and ground-truth accuracy. Your workflow for creating a new task type should be:
>
> (1) Design the Problem: Clearly define the question (e.g., "How many ways can you place the remaining queens?"). Identify its cognitive capability (e.g., N-Queens is a Search problem).
> (2) Implement a "Solver": Write a self-contained Python class or function that can solve the problem for any given parameters. This logic generates the ground-truth answer.
> (3) Implement a "Visualizer": Write a function that generates the visual representation (image) of the problem. This can use matplotlib or HTML as seen in the N-Queens example.
> (4) Parameterize Difficulty: This is the most critical step. Your code must have parameters that control the difficulty (Easy, Medium, Hard).
> Example: In the N-Queens task, we control difficulty by changing the board_size and num_fixed_queens. Placing more fixed queens (e.g., N-2) makes the problem "Easy" because the remaining search space is tiny. Placing fewer (e.g., 1 or 2) makes it "Medium," as it requires a full backtracking search.
> (5) Batch Synthesis: Write a main batch_generate function that loops, calls your solver and visualizer, and saves the output (e.g., image_01.png, image_02.png, etc.) and a corresponding answer file (results.txt).
>
> ## 3. Code Structure Example: N-Queens Task
>
> Your data generator script should follow a similar structure to the one below. This streamlined example demonstrates how the Solver, Visualizer, and Difficulty Control are integrated. (This is a simplified, illustrative version of the nqueens.py script provided.)
> ```
> To ensure diversity across annotators, we organized a brainstorming session in which all annotators jointly proposed a broad set of task ideas before implementation. **For quality control, we performed cross-validation of all generated code: each program was independently reviewed by multiple annotators to verify correctness. Additionally, we designed a suite of test cases for every task type to rigorously validate the solver’s functionality and ensure that the generated ground-truth answers were accurate.**

---

> > ### Author Response · Authors · 2025-11-18
> > **Rebuttal 4**
> >
> > > W4: Although the paper claims to include 1680 tasks, these are derived from only 28 seed task types x 3 difficulty levels x 20 instances per combination, which indicates limited underlying diversity. This structure suggests that most tasks may share similar templates or reasoning structures, differing mainly by parameter changes.
> >
> > We would like to clarify that **our paper consistently states that MMR-VIP encompasses 28 carefully crafted task types, each designed across three difficulty levels, resulting in 1,680 instances** (see Lines 111–112 and Lines 303–304). These instances are not meant to represent 1,680 unique task types, but structured variations within 28 distinct reasoning task families.
> >
> > The 28 task types probe different cognitive-skill dimensions and reasoning structures. Moreover, the three difficulty levels within each task type are not produced through trivial parameter perturbations; instead, they are deliberately designed with distinct intrinsic problem structures, grounded in cognitive load and computational complexity theory.
> >
> > For each task type and difficulty level, we generate 20 instances through controlled parameter variations. **This design ensures that the benchmark remains both robust and statistically reliable because the number of samples is sufficient to reduce randomness and prevent overfitting to any single instance, while still keeping the overall computational cost at a practical level for evaluation.** This approach is also standard practice in reasoning benchmarks [3, 4, 5, 6], where multiple parameterized instances are commonly used to ensure reliable assessment.
> >
> > [3] Yew Ken Chia, Vernon Toh, Deepanway Ghosal, Lidong Bing, Soujanya Poria. PuzzleVQA: Diagnosing Multimodal Reasoning Challenges of Language Models with Abstract Visual Patterns. ACL Findings 2024.
> >
> > [4] Yunfan Shao, Linyang Li, Yichuan Ma, Peiji Li, Demin Song, Qinyuan Cheng, Shimin Li, Xiaonan Li, Pengyu Wang, Qipeng Guo, Hang Yan, Xipeng Qiu, Xuanjing Huang, Dahua Lin. Case2Code: Scalable Synthetic Data for Code Generation. COLING 2025.
> >
> > [5] Kewei Cheng, Jingfeng Yang, Haoming Jiang, Zhengyang Wang, Binxuan Huang, Ruirui Li, Shiyang Li, Zheng Li, Yifan Gao, Xian Li, Bing Yin, Yizhou Sun. Inductive or Deductive? Rethinking the Fundamental Reasoning Abilities of LLMs.
> >
> > [6] Jingqi Tong, Jixin Tang, Hangcheng Li, Yurong Mou, Ming Zhang, Jun Zhao, Yanbo Wen, Fan Song, Jiahao Zhan, Yuyang Lu, Chaoran Tao, Zhiyuan Guo, Jizhou Yu, Tianhao Cheng, Zhiheng Xi, Changhao Jiang, Zhangyue Yin, Yining Zheng, Weifeng Ge, Guanhua Chen, Tao Gui, Xipeng Qiu, Qi Zhang, Xuanjing Huang. Game-RL: Synthesizing Multimodal Verifiable Game Data to Boost VLMs' General Reasoning.
> >
> > > W5: The related work section omits discussion of prior works on visual programming. For example, [7] is also a visual programming benchmark and uses similar code2Task method for data generation. [8] also considers the visual programming setting, in which they synthesize code for visual reasoning. These visual programming work are quite related but not discussed.
> >
> > We thank the reviewer for pointing out these highly relevant prior works in visual programming. **We find [7] to be a particularly interesting and forward-looking benchmark.** It requires models to write programs that control a turtle navigating through a grid to achieve a specified goal, thereby testing a range of skills such as logical reasoning, spatial understanding, planning, mathematical reasoning, and the ability to interpret visual constraints. Notably, [7] also adopts a code-based task synthesis pipeline. **We also find [8] to be highly insightful.** It adopts a neuro-symbolic approach that translates natural language instructions into executable programs for visual reasoning, with each program line capable of invoking off-the-shelf computer vision models, image-processing routines, or Python functions. We apologize for not discussing these works in the original submission and will include a detailed discussion of both [7] and [8] in the revised version.
> >
> > [7] Chao Wen, Jacqueline Staub, Adish Singla. Program Synthesis Benchmark for Visual Programming in XLogoOnline Environment. ACL 2025.
> >
> > [8] Tanmay Gupta, Aniruddha Kembhavi. Visual Programming: Compositional visual reasoning without training. CVPR 2023.

---

> > > ### Author Response · Authors · 2025-11-18
> > > **Rebuttal 5**
> > >
> > > > Q1: In Table 2, why are some open-source models evaluated with T=1 while others include T=3?
> > >
> > > We initially selected Qwen2.5-VL-32B as a relatively stronger and representative open-source model to evaluate its performance under T=3. Following your suggestion, we additionally included Qwen2.5-VL-7B under T=3. Its performance shows only a slight improvement over T=1 and remains close to its CoT results, which further indicates that current open-source models do not benefit substantially from multi-turn code execution and their visual programming capabilities remain limited.
> > >
> > > **Table 1: Qwen2.5-VL-7B Results on MMR-VIP.**
> > > | Setting | Easy   | Mid    | Hard   | Att   | Loc   | Sym    | Geo   | Phy   | Net    | Com    | Sea   | Pla   | Overall |
> > > |---------|--------|--------|--------|-------|-------|--------|-------|-------|--------|--------|-------|-------|---------|
> > > | CoT     | 13.2   | 7.0    | 3.9    | 7.5   | 3.6   | 7.3    | 3.8   | 8.9   | 26.7   | 6.7    | 6.8   | 6.4   | 8.0     |
> > > | T=1     | 7.3    | 5.5    | 1.6    | 2.5   | 2.9   | 7.5    | 2.1   | 2.2   | 17.5   | 21.7   | 5.3   | 1.7   | 4.8  |
> > > | T=3     | 12.7  | 7.7   | 4.8   | 4.5  | 4.1  | 14.0  | 3.3  | 6.7  | 25.8  | 29.2  | 7.7  | 6.7  | 8.4  |
> > >
> > >
> > > > Q2: What exactly defines a native TVP model? Why are such models not evaluated under CoT, T=1, or T=3?
> > >
> > > We refer to native TVP models (e.g., o4-mini, GPT-5) as models that **inherently support multi-turn code execution as part of their built-in inference**. Because of this native design, we cannot reliably constrain them to CoT-only reasoning or restrict their number of tool-use turns (e.g., T=1 or T=3). These models internally decide when and how often to invoke code, and forcing them into a single-turn or no-code mode would fundamentally change their intended reasoning behavior and produce non-comparable results.
> > >
> > > > Q3: How do the authors ensure that the dataset instances represent genuinely distinct reasoning challenges rather than minor parameter variations?
> > >
> > > The 28 task types probe different cognitive-skill dimensions and reasoning structures. Moreover, the three difficulty levels within each task type are not produced through trivial parameter perturbations; instead, they are deliberately designed with distinct intrinsic problem structures, grounded in cognitive load and computational complexity theory.
> > >
> > > For each task type and difficulty level, we generate 20 instances through controlled parameter variations. This design ensures that the benchmark remains both robust and statistically reliable because the number of samples is sufficient to reduce randomness and prevent overfitting to any single instance, while still keeping the overall computational cost at a practical level for evaluation. This approach is also standard practice in reasoning benchmarks [3, 4, 5, 6], where multiple parameterized instances are commonly used to ensure reliable assessment.
> > >
> > > > Q4: How did you come up with these 28 task types? Is it motivated by any existing benchmarks, designed from scratch, or other cases?
> > >
> > > Thank you for the question. The initial inspiration for our benchmark came from the AlgoPuzzleVQA [9], [Toy Theater](https://toytheater.com/), and Leetcode. Building on these inspirations, we further designed the specific task types through internal brainstorming and created multiple difficulty levels for each task type.
> > >
> > > [9] Deepanway Ghosal, Vernon Toh Yan Han, Yew Ken Chia, Soujanya Poria. Are Language Models Puzzle Prodigies? Algorithmic Puzzles Unveil Serious Challenges in Multimodal Reasoning.

---

> > > > ### Author Response · Authors · 2025-11-20
> > > >
> > > > Dear Reviewer 9r1P,
> > > >
> > > > Thank you for your helpful suggestion. We have updated the paper accordingly and added a discussion of relevant visual programming works in the revised version.

---

> ### Author Response · Authors · 2025-11-23
> **Looking forward to your feedback!**
>
> Dear Reviewer 9r1P,
>
> Thank you once again for your valuable feedback. We have added further explanations and clarifications to the paper based on your comments. Since the discussion phase has already begun, we would greatly appreciate knowing whether our responses have addressed your concerns. Your insights are extremely valuable to us, and we are very willing to address any remaining issues to further improve the work.

---

> ### Author Response · Authors · 2025-11-27
> **Looking forward to further discussion with you.**
>
> Dear Reviewer 9r1P,
>
> Thank you once again for your valuable feedback. We have added further explanations and clarifications to the paper based on your comments. We sincerely hope to clarify the misunderstandings and address any concerns you may have. We have introduced the design principles behind the difficulty ladder and cognitive skill dimensions, added more details on benchmark construction, and included a discussion of prior works on visual programming in the related work section. Your insights are extremely valuable to us, and we would be very grateful to hear whether there is any possibility of improving the score at this stage.

---

### Official Review · Reviewer_GxU2 · 2025-10-31

**Soundness:** 3
**Presentation:** 3
**Contribution:** 3
**Rating:** 6
**Confidence:** 3

**Summary:**

This paper proposes a novel paradigm termed **Thinking with Visual Programming (TVP)**. Unlike traditional *Chain-of-Thought (CoT)* or *Thinking with Images (TWI)* frameworks that rely on static visual tools (e.g., crop, zoom, rotate), TVP allows models to **generate, execute, and verify visual and textual code** during reasoning.

The authors introduce **MMR-VIP**, a comprehensive benchmark built on *Visual Impossible Problems* that require agentic reasoning with an external code executor. Tasks are structured along two orthogonal axes:

1. **Difficulty Ladder**—from perception-based “easy” problems in P, to “medium” problems solvable with programming, to “hard” NP-hard-style tasks.
2. **Cognitive Skills**—*Perception*, *Abstraction*, and *Optimization*.

Experiments across commercial, open-source, and native TVP models show that **multi-turn visual programming markedly improves medium-difficulty performance**, with GPT-5 achieving the best overall accuracy (38.2%), while humans remain at 53.6%. The results reveal TVP’s promise in scaling multimodal reasoning depth and highlight abstraction as the main bottleneck.

**Strengths:**

* **Comprehensive benchmark.** MMR-VIP spans 1,680 instances and 28 task types with fine-grained coverage over cognitive skills and complexity tiers.
* **Strong empirical evaluation.** Results across open-source and proprietary models clearly illustrate the scalability and difficulty ladder effects.
* **Clear cognitive framing.** The decomposition into perception, abstraction, and optimization provides a cognitively interpretable lens for diagnosing MLLMs.
* **Reproducibility and ethical standards.** All tasks are programmatically generated, with scripts and evaluation environments promised for release.

**Weaknesses:**

* **Abstraction bottleneck insufficiently analyzed.** While results emphasize abstraction as the limiting factor, the paper provides limited qualitative insight into *why* models fail at symbolic or physical abstraction. Including more introspective analyses (e.g., code trace visualization, abstraction failure typology) would clarify the underlying cognitive gap.
* **Interpretability of TVP reasoning remains underexplored.** Although multi-turn reasoning trajectories are central to the paper’s thesis, few qualitative examples are shown. Representative reasoning chains, success/failure visualizations, or interpretable program outputs would make the paradigm’s dynamics clearer.
* **Limited discussion on computational cost and efficiency.** TVP introduces code execution into the reasoning loop, potentially increasing inference latency and token consumption (as suggested by Figure 4). A deeper discussion on runtime–accuracy trade-offs or efficiency optimizations would be valuable.

**Questions:**

1. How robust is TVP to **execution noise or syntax errors**? Are erroneous interpreter outputs filtered or directly propagated into subsequent reasoning steps?
2. The results indicate that **abstraction remains the major bottleneck**. Do the authors have additional insights on how this skill could be improved—e.g., through reinforcement learning over code-execution trajectories, hierarchical reasoning supervision, or curriculum-based abstraction tasks?
3. **Figure 1(3)** shows substantial reported improvements under TVP, yet the magnitude of the plotted gains seems inconsistent with the y-axis scale.

---

> ### Author Response · Authors · 2025-11-18
> **Rebuttal 1**
>
> Thank you for your thoughtful and detailed feedback. We sincerely hope that our response has effectively addressed your concerns.
>
> > W1: Abstraction bottleneck insufficiently analyzed. While results emphasize abstraction as the limiting factor, the paper provides limited qualitative insight into why models fail at symbolic or physical abstraction. Including more introspective analyses (e.g., code trace visualization, abstraction failure typology) would clarify the underlying cognitive gap.
>
> Thank you for the valuable suggestion. As shown in Appendix F (Error Analysis), we provide a concrete example of the Algorithmic Modeling Error. As illustrated in Figure 43, the model correctly understood the question and extracted the relevant information, but it introduced an incorrect physical constraint by limiting the ball’s horizontal movement to 20 units. This represents a clear **abstraction failure**: the model does not struggle with perception or execution, but rather with constructing accurate symbolic abstractions of the underlying physical rules. **We fully agree with the reviewer’s observation, and following this suggestion, we have added additional abstraction failure analyses in Figures 44 and 45 in the revised paper.**
>
> 1. **Physical Abstraction Failure in Dynamic Simulation**: As shown in Figure 44, the model correctly identified the height and width of the steps and recognized the energy loss during the collision bounce. However, it failed to abstract the underlying physical rules governing the scenario, leading to two major errors. First, it did not correctly formulate or implement the collision-detection mechanism, causing the ball to pass through a region that should have been recognized as a solid step. Second, instead of inferring the initial velocity from the provided short trajectory, the model arbitrarily assigned an initial velocity based on intuition, producing an incorrect simulation outcome. These mistakes reflect a clear breakdown in symbolic and physical abstraction rather than perception.
>
> 2. **Geometric Abstraction Failure in 3D Reconstruction**: As shown in Figure 45, the model attempts to estimate the volume of a triangular prism placed on a checkerboard. It locates several checkerboard corner points and attempts to reason about the positional relationships between the prism’s vertices and the grid. However, the final volume estimate is highly inaccurate. A correct solution would require detecting all checkerboard vertices, inferring the 3D–2D projection transformation matrix, reconstructing the prism’s 3D coordinates from their 2D projections, and then computing the volume. Although the model can perceive local geometric features, it fails to abstract the underlying 3D geometric principles and coordinate-transformation rules needed to solve the task.
>
> > W2: Interpretability of TVP reasoning remains underexplored. Although multi-turn reasoning trajectories are central to the paper’s thesis, few qualitative examples are shown. Representative reasoning chains, success/failure visualizations, or interpretable program outputs would make the paradigm’s dynamics clearer.
>
> **As shown in Appendix G (Case Study), we provide several representative examples that analyze the performance of CoT, single-turn TVP, and multi-turn TVP**. These case studies include qualitative reasoning trajectories, success and failure visualizations, and interpretable program outputs, which together help illustrate the dynamics of the TVP paradigm.
>
> > W3: Limited discussion on computational cost and efficiency. TVP introduces code execution into the reasoning loop, potentially increasing inference latency and token consumption (as suggested by Figure 4). A deeper discussion on runtime–accuracy trade-offs or efficiency optimizations would be valuable.
>
> We agree that computational cost is an important aspect to consider. **We only reported token consumption in Figure 4 because it is challenging to make a fair comparison of runtime between CoT and TVP: the execution time of code can vary substantially across different hardware and environments, making direct runtime measurements incomparable.** We also acknowledge that code execution is relatively time-consuming. Therefore, we believe that enabling the model to adaptively decide whether to invoke code based on the difficulty of the problem represents a promising direction for improving efficiency. We will clarify this point and expand the discussion in the revised version.

---

> > ### Author Response · Authors · 2025-11-18
> > **Rebuttal 2**
> >
> > > Q1: How robust is TVP to execution noise or syntax errors? Are erroneous interpreter outputs filtered or directly propagated into subsequent reasoning steps?
> >
> > Thank you for the question. **As shown in Appendix G, erroneous interpreter outputs are returned directly to the model rather than being filtered. As analyzed in Section 4.3.1, we observe different effects depending on the number of TVP iterations.** For single-turn TVP, erroneous outputs tend to **degrade** performance because the model has no opportunity to correct the faulty code. In contrast, for multi-turn TVP, these error messages are often **beneficial**: the model can use them as feedback signals to iteratively revise its code, leading to improved correctness over multiple reasoning steps.
> >
> >
> > > Q2: The results indicate that abstraction remains the major bottleneck. Do the authors have additional insights on how this skill could be improved—e.g., through reinforcement learning over code-execution trajectories, hierarchical reasoning supervision, or curriculum-based abstraction tasks?
> >
> > Thank you for the thoughtful question. We agree that improving abstraction is crucial. We believe that the first step is to equip the model with visual abstraction skills through supervised fine-tuning, enabling it to internalize the intuition of approaching multimodal tasks through code-based solutions. This provides the necessary foundation for stronger visual programming.
> >
> > Building upon this foundation, multi-turn reinforcement learning over code-execution trajectories becomes a promising next step. By interacting with the interpreter and iteratively refining its code, the model can gradually develop more reliable TVP capabilities.
> >
> > > Q3: Figure 1(3) shows substantial reported improvements under TVP, yet the magnitude of the plotted gains seems inconsistent with the y-axis scale.
> >
> > Thank you for pointing this out. **The intention of Figure 1(3) is to illustrate the relative performance improvements achieved by TVP rather than absolute score differences.**

---

> > > ### Author Response · Authors · 2025-11-21
> > > **Runtime Statistics**
> > >
> > > Dear Reviewer GxU2,
> > >
> > > Following your suggestion, we evaluated the runtime of the code generated by Gemini-2.5-Flash and Claude-Sonnet-4. All experiments were conducted on a dual-socket Intel Xeon E5-2680 v4 server (56 threads, 2.4 GHz base / 3.3 GHz turbo, 70 MB L3 cache) to ensure consistent and controlled execution conditions.
> > >
> > > Below we report the average, minimum, and maximum execution time:
> > >
> > > **Runtime Statistics of Gemini-2.5-Flash.**
> > >
> > > | Metric      | Value (s) |
> > > |-------------|-----------|
> > > | Avg Runtime | 1.3406      |
> > > | Min Runtime | 0.0248       |
> > > | Max Runtime | 60.1643      |
> > >
> > > **Runtime Statistics of Claude-Sonnet-4.**
> > >
> > > | Metric      | Value (s) |
> > > |-------------|-----------|
> > > | Avg Runtime | 2.0226       |
> > > | Min Runtime | 0.0274       |
> > > | Max Runtime | 60.1436      |

---

> > > > ### Comment · Reviewer_GxU2 · 2025-11-24
> > > > **Thanks for the response**
> > > >
> > > > The author response has addressed most of my concerns. I will retain my score recommending acceptance.

---

> > > > > ### Author Response · Authors · 2025-11-25
> > > > > **Appreciate your recommendation for acceptance**
> > > > >
> > > > > Dear Reviewer GxU2,
> > > > >
> > > > > Thank you very much for your thoughtful and insightful feedback. We are pleased that our responses were able to address your concerns. Your suggestions have been highly valuable and have helped guide us toward a more comprehensive revision. We truly appreciate your recommendation for acceptance and sincerely thank you for your support.

---

### Official Review · Reviewer_zm6s · 2025-11-01

**Soundness:** 3
**Presentation:** 3
**Contribution:** 3
**Rating:** 6
**Confidence:** 4

**Summary:**

The paper proposes Thinking with Visual Programming (TVP), where models generate and run code, including visual operations, as part of reasoning. It introduces MMR-VIP, a benchmark of “Visual Impossible Problems” across different difficulty levels and cognitive skills. Experiments show that iterative code execution improves medium-level performance, though models still struggle with abstraction.

**Strengths:**

-	paper introduces a new benchmark to facilitate the development of VLMs.
-	It reveals that existing VLMs struggle with visual programming tasks.
-	It proposes a data generation framework for fine-tuning.

**Weaknesses:**

-	I think 28 task types are still somewhat limited. It would be better if the benchmark could be scaled up through more automated task generation methods to cover a broader and more diverse set of problems.
-	Although GPT-5 and o4 are labeled as native TVP models, I think it would be useful to also evaluate them under the same external-tool setup used for other models (e.g., through prompts that block internal executors). That would make the comparison fairer and help understand how much of the advantage comes from the model itself versus the interface.
-	The “Human” results lack participant details such as background, age, or expertise. For a paper of this type, similar to what psychology or cognitive science works do, including such information would make the human baseline more credible.
-	The paper could discuss more related benchmarks, for example Humaneval-V [1], to clarify how MMR-VIP differs from or complements existing multimodal reasoning datasets. It is recommended to add a discussion on visual reasoning benchmarks in the related work section.

[1] https://arxiv.org/pdf/2410.12381

**Questions:**

Please see weakness.

---

> ### Author Response · Authors · 2025-11-18
> **Rebuttal 1**
>
> Thank you for your thoughtful and detailed feedback. We sincerely hope that our response has effectively addressed your concerns.
>
> > W1: It would be better if the benchmark could be scaled up through more automated task generation methods to cover a broader and more diverse set of problems.
>
> We adopted a strategy of manually writing the data generation code to ensure the quality of the synthesized dataset. **This approach allows us to more precisely control the accuracy, difficulty, and originality of the tasks.** However, it also limits the overall dataset size and the breadth of problem coverage.
>
> We appreciate the reviewer’s suggestion regarding scaling up through more automated task generation. **Using our manually constructed tasks as seeds, leveraging LLMs to synthesize additional tasks is indeed a promising direction, as it could rapidly expand both the diversity and volume of tasks and further enrich the benchmark.** At the same time, fully automated generation also introduces important challenges, such as controlling task difficulty, ensuring the correctness of both tasks and reference answers, and maintaining a balanced distribution of task types. We will explore these directions in future work.
>
>
> > W2: Although GPT-5 and o4 are labeled as native TVP models, I think it would be useful to also evaluate them under the same external-tool setup used for other models (e.g., through prompts that block internal executors).
>
> We appreciate this insightful suggestion. We attempted to evaluate GPT-5 under the same setting by explicitly instructing the models in the prompt not to invoke internal executors. However, in practice, **the model often did not follow these instructions, and its behavior remained largely unchanged**. As a result, their performance under this “blocked-executor” setting was almost identical to the results reported in the paper. To avoid drawing conclusions from an uncontrolled setup, we chose not to include these results in Table 2.
>
> > W3: The “Human” results lack participant details such as background, age, or expertise. For a paper of this type, similar to what psychology or cognitive science works do, including such information would make the human baseline more credible.
>
> We apologize for not providing details about the human participants in the original submission. We recruited four participants to solve the tasks, and each participant was allowed to use search engines and code interpreters and was proficient in using these tools during the process. Their background information is summarized in the table below.
>
> | Participant   | Age | Background                         | Expertise                                             |
> |---------------|-----|--------------------------------------|--------------------------------------------------------|
> | Participant 1 | 21  | AI (Undergraduate student) | Proficient in applying mathematical tools to problem solving |
> | Participant 2 | 23  | CS (Graduate student)               | Strong experience in informatics competitions          |
> | Participant 3 | 24  | EE (Graduate student)         | Skilled in software development and programming         |
> | Participant 4 | 27  | CS (Graduate student)               | Moderate experience in informatics competitions         |
>
>
>
> > W4: The paper could discuss more related benchmarks, for example Humaneval-V [1], to clarify how MMR-VIP differs from or complements existing multimodal reasoning datasets. It is recommended to add a discussion on visual reasoning benchmarks in the related work section.
>
> We greatly appreciate the reviewer’s suggestion to discuss additional multimodal reasoning benchmarks such as HumanEval-V. **HumanEval-V is a solid and forward-looking benchmark designed to evaluate complex diagram understanding and visual reasoning abilities in programming contexts.** Its tasks require the model to infer the rules encoded in a diagram and implement a function that applies these rules to pass all test cases. The final output is executable code whose behavior is fully determined by the diagram’s visual content.
>
> The primary goal of MMR-VIP is to evaluate a model’s multimodal reasoning capabilities, where code functions only as an optional tool to enhance the reasoning process rather than as the output target itself. All code generated in MMR-VIP is free-form and intended solely to aid problem solving. We will incorporate a detailed discussion of HumanEval-V and related visual reasoning benchmarks in the revised version.

---

> > ### Author Response · Authors · 2025-11-20
> >
> > Dear Reviewer zm6s,
> >
> > Thank you for your helpful suggestion. We have updated the paper accordingly and added a discussion of the relevant multimodal reasoning benchmarks in the revised version.

---

### Author Response · Authors · 2025-12-03
**Rebuttal Summary for the Area Chair (1/2)**

Dear AC,

We sincerely regret the unexpected termination of the discussion period and greatly appreciate your dedication throughout the review process. Given the current situation, we fully understand that it may be helpful for you to quickly become familiar with the key points of our work and the progress of the rebuttal. **To better assist your evaluation, we have prepared a clear and concise summary of the entire discussion.** Our summary consists of three parts: **an overview of the discussion process, the strengths identified by the reviewers, and the common concerns they raised**.

### **An Overview of the Discussion Process**

**During the discussion, we were grateful to receive responses from Reviewer GxU2. We are very pleased that our rebuttal addressed the reviewer’s concerns, and that Reviewer GxU2 decided to retain the 6/10 score recommending acceptance.** Unfortunately, due to the unexpected termination of the discussion period, we were unable to receive responses from the remaining reviewers. **Nevertheless, given that their concerns were similar in nature, we believe that our detailed rebuttal would likewise address their questions effectively.**

### **Strengths Highlighted by the Reviewers**

The reviewers highlighted several key contributions of our work:

1. **Comprehensive Benchmark**: MMR-VIP covers 1,680 instances across 28 task types with fine-grained cognitive and complexity coverage, providing a valuable resource for evaluating multimodal LLMs (Reviewers GxU2, 9r1P, TRk7, zm6s).

2. **Clear Cognitive Framing**: The decomposition into perception, abstraction, and optimization offers a cognitively interpretable lens for diagnosing MLLMs, making it highly valuable for evaluating their reasoning capabilities (Reviewers GxU2, 9r1P).

3. **Strong Empirical Evaluation**: Results across both open-source and proprietary models clearly demonstrate the scalability and difficulty-ladder effects, highlighting the deficiencies of current TVP models (Reviewers GxU2, TRk7).

---

> ### Author Response · Authors · 2025-12-03
> **Rebuttal Summary for the Area Chair (2/2)**
>
> ### **Common Concerns Raised by the Reviewers**
>
> We are also very grateful for the concerns raised by the reviewers, which are essential for improving the quality and clarity of our work. The major concerns primarily relate to the theoretical grounding of the benchmark, the need for more thorough qualitative analysis, the details of dataset construction, and the discussion of existing datasets. Below, we provide abbreviated responses that concisely address each of these points.
>
> 1. The theoretical grounding of the benchmark (W1 of Reviewer 9r1P, W2 of Reviewer 9r1P, W2 of Reviewer TRk7): **We would like to clarify that our framework is rigorously grounded in established theories rather than model observation**:
>
> - **Difficulty Ladder:** Defined by **Computational Complexity Theory** and **Cognitive Load**. "Easy" tasks correspond to low-complexity problems in $P$ (solvable via working memory); "Medium" tasks are polynomial-time problems requiring external computation (exceeding working memory); "Hard" tasks conceptually correspond to $NP$-hard problems requiring advanced strategies, for example, innovative algorithms or special optimizations for specific problems.
> - **Cognitive Dimensions:** Our taxonomy aligns with the **Perception-Cognition-Action (PCA)** model in cognitive science. We map *Perception* to sensory processing, *Abstraction* to internal representation/modeling, and *Optimization* to goal-directed action selection.
>
> 2. The need for more thorough qualitative analysis (W1 of Reviewer GxU2, W2 of Reviewer GxU2, W5 of Reviewer TRk7): We would like to clarify that **the original submission already includes an error analysis (Appendix F) and a case study (Appendix G)**, which may not have been fully noticed during the initial review. These sections provide detailed qualitative insights into model behaviors and failure patterns.
>
> 3. The details of dataset construction (W3 of Reviewer 9r1P, W4 of Reviewer 9r1P, W3 of Reviewer TRk7): **To ensure the quality and label correctness of the benchmark, the entire dataset is generated using a Code2Task generation framework.** In this framework, annotators are responsible for writing code that specifies task rules and automatically generates the corresponding images, problems, and answers. As the benchmark is fully code-generated, the focus is primarily on ensuring the correctness of the underlying code logic. As long as the code logic is correct, the generated tasks, images, and answers are inherently reliable and free from labeling errors. In fact, **we have already provided a concise annotation guideline in Appendix C (Page 16).**
>
> 4. The discussion of existing datasets (W4 of Reviewer zm6s, W5 of Reviewer 9r1P): **We have included a discussion of prior works on visual programming in the related work section.**
>
> In summary, we have addressed the concerns raised by Reviewer GxU2. **Although we were unable to receive further responses from the remaining reviewers due to the leak incident, we have provided substantial evidence indicating the potential for a positive reassessment of our work.**
>
> **Once again, we would like to express our deep appreciation for your invaluable role and the significant time you have devoted throughout this process. We also extend our sincere thanks to the reviewers for their meticulous feedback and thoughtful suggestions.**
>
> Best regards,
>
> Authors

---

### Meta-Review · Area_Chair_HKWt · 2025-12-24

**Summary:**

This submission proposes Thinking with Visual Programming (TVP) and introduces the MMR-VIP benchmark (1,680 instances across 28 task types) to evaluate multimodal agentic reasoning with iterative code execution. Reviewers generally found the benchmark framing and empirical results interesting, but raised concerns that reduce confidence in the paper’s positioning and methodological clarity.

The main issues were (1) whether the “difficulty ladder” and the Perception/Abstraction/Optimization taxonomy are intrinsically defined and sufficiently justified, rather than being model- or implementation-dependent, (2) limited underlying diversity given 28 seed task families with parameterized instances, (3) insufficient transparency around benchmark construction and the human baseline reporting, plus related responsible-research details, (4) incomplete related-work coverage on visual programming benchmarks, and (5) limited qualitative evidence and cost/efficiency discussion for TVP trajectories.

Overall, the rebuttal provides clarifications and additional pointers, but **most changes are explanatory rather than presenting new objective evidence that resolves the primary doubts**, so the final recommendation remains Reject.

**Reviewer Concerns:**

**Addressed or partially addressed**:

- Related work omissions: authors acknowledge missing visual programming benchmarks and state they will add discussion.
- Benchmark construction transparency and ethics details: authors clarify that annotators are co-authors, provide a more detailed guideline and describe cross-checking and test cases; they also provide human participant background information for the human baseline. This improves clarity but still leaves some responsible-research details underspecified.
- Qualitative analysis request: authors point to existing appendices and claim added more failure analyses in a revision. This helps, but the reviewer concern was about the strength and representativeness of qualitative evidence in the main narrative.

**Still outstanding:**

- **Difficulty ladder validity**: the rebuttal argues grounding in complexity theory and “cognitive load,” but the operational definitions still appear tied to tool availability, timeouts, and what current systems can practically solve. The response reads more as a rationale than an independently verifiable definition with clear, task-intrinsic criteria and validation.
- **Cognitive skill dimensions**: citing PCA-style framing helps, but the paper still lacks empirical validation that the three axes are distinctive, reliably labeled, and useful beyond an intuitive categorization.
- **Diversity and scaling**: the core limitation remains that the benchmark is built from 28 task families; the rebuttal explains why this is acceptable and standard, but does not provide new evidence that template effects are limited or that coverage is broad enough for the paper’s stronger claims.
- **Fairness of comparisons involving “native TVP” models**: the rebuttal states they cannot reliably disable internal tool use, but this means some comparisons remain difficult to interpret and the paper should be careful about claims that rely on cross-setting fairness.
- **Efficiency discussion**: adding limited runtime stats for a couple models is helpful, but does not fully address the broader runtime and cost tradeoff across settings and models.

**Reviewer Scores:**

**Reviewer zm6s (6)**: Likely unchanged. The rebuttal reasonably addresses the human-baseline detail request and promises expanded related work, but the reviewer’s broader concerns about task-type breadth and fairness of evaluating “native TVP” models under comparable constraints remain only partially resolved.

**Reviewer GxU2 (6)**: Likely unchanged. The authors responded directly on qualitative analysis, robustness to execution errors, and provided limited runtime statistics. These address several questions but do not fundamentally change the evidentiary basis for the main claims.

**Reviewer 9r1P (4)**: Likely unchanged. The primary skepticism concerns whether the difficulty ladder and cognitive axes are intrinsically defined and validated, plus concerns about diversity and construction transparency. The rebuttal mainly provides conceptual justification and additional description rather than objective evidence.

**Reviewer TRk7 (4)**: Likely unchanged. The rebuttal corrects at least one factual misunderstanding and points to existing appendices, but does not materially add new evidence that would warrant a score increase, and the review’s substantive concerns remain only broadly addressed.

---

### Decision · Program_Chairs · 2026-01-26

Reject